# Online Uniform Sampling: Randomized Learning-Augmented Approximation Algorithms with Application to Digital Health

## Abstract

Motivated by applications in digital health, this work studies the novel problem of *online uniform sampling* (OUS), where the goal is to distribute a sampling budget uniformly across *unknown* decision times. In the OUS problem, the algorithm is given a budget $b$ and a time horizon $T$, and an adversary then chooses a value $\tau^* \in [b, T]$, which is revealed to the algorithm online. At each decision time $i \in [\tau^*]$, the algorithm must determine a sampling probability that maximizes the budget spent throughout the horizon, respecting budget constraint $b$, while achieving as uniform a distribution as possible over $\tau^*$. We present the first randomized algorithm designed for this problem and subsequently extend it to incorporate learning augmentation. We provide *worst-case* approximation guarantees for both algorithms, and illustrate the utility of the algorithms through both synthetic experiments and a real-world case study involving the HeartSteps mobile application. Our numerical results show strong empirical *average* performance of our proposed randomized algorithms against previously proposed heuristic solutions.

## 1. Introduction

The problem of *online uniform sampling* (OUS) is motivated by applications in digital health, where administering interventions at inappropriate times, such as when users are not at risk,[1] can significantly increase mental burden and hinder engagement with digital interventions (Li et al., 2020; Nahum-Shani et al., 2018; Wen et al., 2017; McConnell et al., 2017; Mann & Robinson, 2009). Existing studies (Heckman et al., 2015; Klasnja et al., 2008; Dim-

itrijević et al., 1972) show excessive digital interventions can heighten user fatigue, suggesting a threshold beyond which intervention effectiveness declines. A strategy rooted in the ecological momentary assessment (EMA) literature and proven effective in mitigating user fatigue *involves allocating a fixed and limited budget for treatments delivered to the patient and delivering them with a uniform distribution across all risk times* (e.g., Liao et al. 2018; Dennis et al. 2015; Rathbun et al. 2013; Scott et al. 2017a;b; Shiffman et al. 2008; Stone et al. 2007). However, this strategy is challenging because the true number of risk times is unknown, inspiring the OUS problem.

**Contributions** Our contributions in this paper are two-fold. **First**, we formulate the common OUS problem in digital health as an online optimization problem and provide randomized algorithms that perform well in practice with *competitive ratio* guarantees. The competitive ratio measures the performance of an online algorithm against an offline clairvoyant benchmark, assuming the unknown parameter is revealed to the clairvoyant in advance. These guarantees are inherently conservative: 1) no online algorithm can achieve the same performance as the clairvoyant in practice (i.e., a competitive ratio of 1 is unattainable in OUS), and 2) they hold across *all* problem instances or sample paths (i.e., they are worst-case guarantees). Consequently, online approximation algorithms may exhibit conservative behavior. To address this, we numerically illustrate the practicality of our algorithm, demonstrating that they outperform naive benchmarks on average.

**Second**, we extend our algorithm to the practical setting where a confidence interval *containing* the true risk time is provided, potentially through a valid statistical inference procedure. We conduct the competitive ratio analysis for our proposed learning-augmented approximation algorithm, demonstrating its *consistency* in the strong sense—optimal performance is achieved when the confidence interval width is zero—and *robustness*—the learning-augmented algorithm performs no worse than the non-learning augmented counterpart. Our findings indicate that, in almost all tested scenarios, the randomized learning-augmented algorithm outperforms its non-learning augmented counterpart.

**Outline** In Section 2, we formalize the OUS problem. We in-

---

[1]Anonymous Institution, Anonymous City, Anonymous Region, Anonymous Country. Correspondence to: Anonymous Author <anon.email@domain.com>.

Preliminary work. Under review by the International Conference on Machine Learning (ICML). Do not distribute.

[1]Risk times are when the patient is susceptible to a negative event, such as smoking relapse.

troduce our randomized algorithm without learning augmentation in Section 3. This algorithm is segmented into three distinct cases based on the horizon length to budget ratio, with a competitive ratio established for each. In Section 4, we develop a learning-augmented algorithm that integrates a prediction interval and provide theoretical justification for its effectiveness. The efficacy of these algorithms is first assessed through synthetic experiments, followed by their application to real-world data in Section 5.

### 1.1. Related Work

**Online Uniform Sampling** Existing methodologies, primarily sourced from the EMA literature, focus on delivering interventions through the form of mobile self-report requests over a fixed time horizon. These approaches are constrained by budget and uniformity considerations to minimize user burden and ensure accurate reflection of user conditions across diverse contexts (Dennis et al., 2015; Rathbun et al., 2013; Scott et al., 2017a;b). In this work, we permit intervention only when users are *at risk*, leading to an unknown horizon length. This introduces a significant challenge in balancing the allocation of a limited budget with the need to maintain uniformity in intervention delivery. To address this issue, Liao et al. (2018) developed a heuristic algorithm, but its performance depends heavily on the accuracy of the predicted number of risk times. When the prediction is inaccurate, the algorithm lacks theoretical guarantees, highlighting the need for a more robust algorithm design.

**Multi-option Ski-rental Problem** Our work closely relates to the *multi-option ski-rental* (MOSR) problem (Zhang et al., 2011; Shin et al., 2023), where the number of snowy days is unknown. Customers have multiple ski rental options, differing in cost and duration. The goal is to minimize costs while ensuring ski availability on snowy days. Shin et al. (2023) introduced a randomized algorithm for MOSR, with a tight $e$-competitive ratio. A random variable $B$ is introduced as a proxy for the unknown true horizon $T$. $B$ is initialized to $\alpha$, following a density function $1/\alpha$ within $[1, e)$. The algorithm iteratively solves an optimization problem to identify an optimal set of rental options within budget $B$, maximizing day coverage. Customers sequentially utilize the options until depletion, at which point $B$ is increased by a factor of $e$, and the process is repeated.

Our work builds upon Shin et al. (2023), leveraging the same randomized algorithmic idea. However, our problem setting is *significantly different* from that of MOSR. In particular, instead of having discrete ski-rental options, at each decision time, the algorithm needs to decide on the sampling probability, which is continuous in nature. Further, in our problem, the sum of the sampling probability cannot exceed a predefined budget, while such constraints do not exist in MOSR. Our problem additionally has a uniformity consideration.

**Learning-Augmented Online Algorithms** Many online algorithms incorporate black-box point predictions on the unknown parameters to improve their worst-case guarantees (Purohit et al., 2018; Bamas et al., 2020; Wei & Zhang, 2020; Jin & Ma, 2022). The confidence of these point estimates is often represented by a single parameter, with a higher value indicating more accurate predictions. When the confidence is low, most work do not guarantee that the learning-augmented algorithm will perform no worse than the non-learning counterpart (Bamas et al., 2020). In practice, prediction confidence intervals, rather than point estimates, are often generated using valid statistical inference methods. A wider confidence interval typically indicates less informative predictions (Shafer & Vovk, 2008). Im et al. (2021) consider the setting where the prediction provides a range of values for key parameters in the online knapsack problem. However, their deterministic solution cannot be directly extended to our setting, as the number of risk times in OUS is stochastic. We introduce the first integration of confidence intervals into randomized algorithms for OUS. This integration enables our proposed algorithms to surpass the performance of their non-learning counterpart, even with a wide confidence interval.

## 2. Problem Framework

In the context of digital interventions, we define the OUS problem as presented by Liao et al. (2018). Let $T$ denote the total number of decision points within a decision period (e.g., within a day). At any given time $t \in [1, T]$ in each decision period, patients encounter binary risk levels[2] (determined by data from wearable devices), indicating whether the patient is likely to experience an adverse event, such as relapse to smoking. The distribution of risk levels is allowed to change *arbitrarily* across decision periods since treatments may influence and reduce subsequent risk.

Let $\tau^*$ be the *unknown true* number of risk times that a patient experiences in a decision period. Note that $\tau^*$ is stochastic and is revealed *only* at the end of the horizon $T$, corresponding to the last decision time in the decision period. We define $p_i \in (0, 1)$ to be the treatment probability at time $i \in [\tau^*]$. We preclude the possibility that $p_i = 0$ or $p_i = 1$ to facilitate after-study inference (Boruvka et al., 2018; Zhou et al., 2023; Kallus & Zhou, 2022).

The algorithm is provided with a *soft* budget of $b$, representing the total *expected* number of interventions allowed to be delivered within each decision period. We assume $\tau^* > b$ as evidenced in practice (Liao et al., 2018). At each decision time $i$, the algorithm decides the intervention probability $p_i$. The objectives of the OUS problem (Liao et al., 2018; Den-

---

[2]When multiple risk levels are present, the problem naturally decomposes into independent subproblems for each risk level, see more details in Appendix A.

nis et al., 2015; Rathbun et al., 2013; Scott et al., 2017a;b; Shiffman et al., 2008; Stone et al., 2007) are to 1) assign the intervention probabilities $\{p_i\}_{i\in[\tau^*]}$ as uniform as possible across risk times, and 2) maximize the sum of intervention probabilities across risk times while adhering to the budget constraint $b$.

Abstractly, in the OUS problem, the algorithm is given a budget $b$ and a time horizon $T$, and an adversary then chooses a value $\tau^* \in [b, T]$, which is revealed to the algorithm online. At each decision time $i \in [\tau^*]$, the algorithm must determine a sampling probability that maximizes the budget spent throughout the horizon, respecting the budget constraint $b$, while achieving as uniform a distribution as possible over $\tau^*$.

*Without additional information on* $\tau^*$, the two objectives compete with each other. A naive solution to fulfill the first objective is to set $p_i = b/T, i \in [\tau^*]$, which, however, fails to maximize the sum of intervention probabilities. Conversely, if we set $p_i$ to be a large constant value, there is a risk of depleting the budget before the end of the horizon, thus failing to achieve the uniformity objective. Therefore, the optimality of the two objectives cannot be simultaneously achieved without additional information on $\tau^*$. Liao et al. (2018) provided a heuristic algorithm for OUS given a point estimate of $\tau^*$. The algorithm's performance is significantly influenced by the accuracy of this forecast. In this work, we introduce randomized algorithms for OUS with robust worst-case guarantees, considering settings both with and without learning augmentation.

### 2.1. OUS as An Online Optimization Problem

In this section, we formulate OUS as an online optimization problem, where the objective function provides a uniform way of comparing the performance of different approximation algorithms, and the constraint defines the set of feasible solutions.

Specifically, we aim to find a sequence of treatment probability assignments $\{p_i\}_{i\in[\tau^*]}$ that achieves the following two objectives:

1. Maximizes the sum of treatment probabilities across risk times, subject to the "soft" budget $b$;
2. Penalizes changes in treatment probabilities within each risk level.

Formally, the OUS problem can be expressed using the following optimization problem:

$$\left\{ \max \sum_i^{\tau^*} p_i - \frac{1}{\tau^*} \ln\left(\frac{\max_{i\in[\tau^*]} p_i}{\min_{i\in[\tau^*]} p_i}\right) : \\ \mathbb{E}\left[\sum_{i=1}^{\tau^*} p_i\right] \le b, p_i \in (0,1), \forall i \in [\tau^*]. \right\} \quad (1)$$

where the expectation, $\mathbb{E}$, in the budget constraint is taken over the randomness in the algorithm. This budget constraint is "soft" in the sense that if we have multiple decision periods (which is the case in digital health), we should satisfy the budget constraint in expectation.

*Remark* 2.1. Notably, the purpose of formulating the optimization problem is not to solve it optimally, but rather to provide a feasible solution without knowledge of the unknown $\tau^*$. Rather than setting uniformity as a constraint, we incorporate it into the design of our approximation algorithms. By including uniformity as a penalty term in the objective function, represented by:

$$\frac{1}{\tau^*} \ln\left(\frac{\max_{i\in[\tau^*]} p_i}{\min_{i\in[\tau^*]} p_i}\right), \quad (2)$$

we can directly compare the overall performance of different online approximation algorithms, including how well they achieve uniformity, by comparing their objective function values.

The choice of the penalty term (2) is inspired by the entropy change concept from thermodynamics (Smith, 1950). This choice is not unique but it has several nice properties: a) it equals to 0 if and only if $\{p_i\}_{i\in[\tau^*]}$ are identical, b) it increases with the maximum difference in $\{p_i\}_{i\in[\tau^*]}$, and c) it tends towards infinity as the value of $p_i$ approaches to zero, penalizing scenarios where the expected budget is depleted before the horizon ends. We note that one can replace the term $1/\tau^*$ in the penalty by a tuning parameter $\sigma$, which controls the strength of the penalty, as discussed in Remarks 3.3 and 4.3.[3] Finally, we highlight that KL divergence cannot be used here to impose uniformity (see detailed discussion in Appendix B).

### 2.2. Offline Clairvoyant and Competitive Ratio

In the *offline clairvoyant* benchmark, the clairvoyant possesses knowledge of $\tau^*$. When provided with this value, the optimal solution to Problem (1) is to set $p_i = b/\tau^*$. Consequently, the optimal value of the objective function in Problem (1) is $\mathrm{OPT}(\tau^*) = b$. Importantly, in practice, no online algorithm can attain $\mathrm{OPT}(\tau^*)$ as the offline clairvoyant benchmark serves as an upper bound on the best achievable performance for any *online* algorithm without knowledge of $\tau^*$. Let SOL be the objective value of Problem (1) achieved by a *randomized online* algorithm, we say that

**Definition 2.2** ($\gamma$-competitive)**.** An algorithm is $\gamma$-*competitive* if $\mathbb{E}[\mathrm{SOL}] \ge \gamma \cdot \mathrm{OPT}(\tau^*)$.

*Remark* 2.3. First, we emphasize that the expectation in

---

[3]Since the current design of our algorithms does not explicitly account for the form of the penalty term, the penalty (2) could also be replaced by any other suitable functions, with performance re-evaluated under the modified objective function.

Definition 2.2 is taken *only* over the randomness of the algorithm. Second, we note that if the competitive ratio is provided, it holds in expectation for every feasible $\tau^* \in [b, T]$. This implies that the competitive ratio serves as a worst-case guarantee: in any OUS instance, as long as the budget $b$ and the maximum horizon length $T$ remain fixed across decision periods, we can expect to meet the budget and achieve the stated competitive ratio, regardless of the specific realization of $\tau^*$ in each decision period.

The key difficulty in solving Problem (1) in the online setting arises due to the unknown nature of $\tau^*$. In Section 3, we introduce the first approximation algorithm for the OUS problem.

### 2.3. With Learning Augmentation

In the *learning-augmented* setting, we are additionally provided with a prediction confidence interval $[L, U]$, generated by a valid statistical procedure, that contains the unknown *true* $\tau^*$ with high probability. A wider confidence interval reflects lower prediction quality. For simplicity, we assume $\tau^*$ lies within the interval, though our results generalize to cases where it is contained with high probability.

To evaluate the performance of the learning-augmented algorithm in the presence of a prediction confidence interval, we extend the standard consistency-robustness analysis from the prior literature (Lykouris & Vassilvtiskii, 2018; Purohit et al., 2018; Bamas et al., 2020; Shin et al., 2023). Specifically, an algorithm is said to be $\lambda$-*consistent* if it achieves $\mathbb{E}[\text{SOL}] \geq \lambda \cdot \text{OPT}(\tau^*)$ when the prediction is perfect, i.e., when $L = U$, indicating a zero-length interval.[4] This aligns with the standard definition where the prediction is accurate (Shin et al., 2023). Conversely, an algorithm is $\rho$-*robust* if it satisfies $\mathbb{E}[\text{SOL}] \geq \rho \cdot \text{OPT}(\tau^*)$ regardless of the width of the prediction interval $[L, U]$, corresponding to the previous definition where the prediction can be arbitrarily inaccurate.

In Section 4, we show that our proposed learning-augmented algorithm is 1-consistent, achieving the optimal solution when the interval width is zero. Moreover, the competitive ratio of our learning-augmented algorithm closely matches that of the non-learning augmented counterpart, even when the prediction quality deteriorates. To the best of our knowledge, this is the first work that provide a 1-consistency guarantee on learning-augmented algorithms, after careful engineering of the algorithms.

## 3. Randomized Algorithm

In this section, we introduce our randomized algorithm, Algorithm 1, designed for the OUS problem *without* learning

---

[4]Similar to Definition 2.2, the expectation is taken over the randomness in the algorithm.

augmentation. This algorithm is inspired by the randomized algorithm proposed by Shin et al. (2023) for the MOSR problem. Due to the significant differences in problem setup outlined in Section 1.1, the design of our algorithm requires 1) imposing a discrete structure on the sampling probabilities to account for uniformity considerations, making the analysis of the algorithm more tractable, and 2) explicitly addressing the finite horizon length and budget constraint, ensuring that the randomized algorithm does not exceed the budget in expectation.

---

**Algorithm 1** Randomized Online Algorithm

1: **Input:** $T$, $b$
2: **Initialize:** $j = 1$, we sample $\alpha \in [b, be]$ from a distribution with p.d.f. $f(\alpha) = 1/\alpha$, and initialize $\tilde{\tau} = \alpha$
3: **for** $i = 1, ..., \tau^*$ **do**
4:     We calculate:
$$\text{Int}(\tilde{\tau}) = \begin{cases} \lfloor \tilde{\tau} \rfloor & w.p. & \lceil \tilde{\tau} \rceil - \tilde{\tau} \\ \lceil \tilde{\tau} \rceil & w.p. & \tilde{\tau} - \lfloor \tilde{\tau} \rfloor \end{cases}$$
5:     **if** $T \leq be$ **then**
6:         Update $\tilde{\tau}$ and set $p_i$ using **Subroutine** 1
7:     **else if** $be < T \leq be^2$ **then**
8:         Update $\tilde{\tau}$ and set $p_i$ using **Subroutine** 2
9:     **else**
10:        Update $\tilde{\tau}, b$ and set $p_i$ using **Subroutine** 3
11:    **end if**
12:    Output treatment probability $p_i$
13: **end for**

---

The proposed algorithm, Algorithm 1, provides a feasible solution to Problem (1). At its core, our algorithm assigns the sampling probabilities in a monotonically non-increasing fashion over time. To accommodate varying practical scenarios where the budget-to-horizon ratio differs across applications, we designed specialized approximation algorithms for three possible scenarios: 1) $T \leq be$ (**Subroutine** 1), 2) $be < T \leq be^2$ (**Subroutine** 2), and 3) $T > be^2$ (**Subroutine** 3).

We maintain a running "guess" of $\tau^*$, denoted by $\tilde{\tau}$. We initialize $\tilde{\tau}$ to be $\alpha$, where $\alpha \sim [b, b \cdot e]$ with density $1/\alpha$, and $e$ represents the Euler's number. If the current number of risk times $i$ is within our running guess $\tilde{\tau}$, then we do not change the current sampling assignment probability. Otherwise, we update $\tilde{\tau}$ as $\tilde{\tau} = \tilde{\tau}e$ and update the sampling probability according to Algorithm 1, depending on the length of the horizon $T$ relative to $b$. The random draw $\tilde{\tau}$ controls not only the value of the sampling probability but also the duration of each stage. Once the algorithm reaches $\tilde{\tau}$, it transitions to the next stage, resulting in a stage-wise constant probability sequence.

We first show the feasibility of our proposed solution, i.e.,

**Subroutine 1** $(i, b, \tilde{\tau}, T, \text{Int}(\tilde{\tau}))$

1: **if** $i > \text{Int}(\tilde{\tau})$ **then**
2:     $\tilde{\tau} = \tilde{\tau} e$
3: **end if**
4: $p_i = \frac{b}{\min(T, \tilde{\tau}(e-1))}$

---

**Subroutine 2** $(i, b, \tilde{\tau}, \text{Int}(\tilde{\tau}))$

1: **if** $i > \text{Int}(\tilde{\tau})$ **then**
2:     $j = j + 1, \tilde{\tau} = \tilde{\tau} e$
3: **end if**
4: **if** $j \geq 3$ **then**
5:     $p_i = \frac{b}{\tilde{\tau} e}$
6: **else**
7:     $p_i = \frac{b}{\tilde{\tau}(e-1)}$
8: **end if**

---

**Subroutine 3** $(i, b, \tilde{\tau}, \text{Int}(\tilde{\tau}))$

1: **if** $i > \text{Int}(\tilde{\tau})$ **then**
2:     $j = j + 1, \tilde{\tau} = \tilde{\tau} e$
3:     **if** $j \geq 3$ **then**
4:         $b = b(1 - \frac{1}{e})$
5:     **end if**
6: **end if**
7: $p_i = \frac{b}{\tilde{\tau} e}$

---

the sampling probabilities outputted from Algorithm 1 satisfies the budget constraint in Problem (1):

**Lemma 3.1.** *Let $p_i^{A1}$ be the probability returned by Algorithm 1 at risk time $i \in [\tau^*]$. This solution always satisfies the budget constraint in expectation, i.e., $\mathbb{E}\left[\sum_{i=1}^{\tau^*} p_i^{A1}\right] \leq b$, where the expectation is taken over the randomness of the algorithm.*

The proof of Lemma 3.1 is included in Appendix C.1. Next, by leveraging the monotonically non-increasing nature of the sampling probabilities, the objective in Problem (1) simplifies to

$$\max \sum_{i=1}^{\tau^*} p_i - \frac{1}{\tau^*} \ln\left(\frac{p_1}{p_{\tau^*}}\right). \tag{3}$$

Using Equation (3), we compute the competitive ratio of Algorithm 1:

**Theorem 3.2.** *Algorithm 1 is $\mathcal{X}(T)$-competitive, where $\mathcal{X}$ is defined as follows:*

$$\mathcal{X}(T) := \begin{cases} \frac{1}{e}\left(\ln(e-1) + \frac{1}{e-1}\right) & \text{if} \quad T \leq be, \\ \frac{1}{e} & \text{if} \quad be < T \leq be^2, \\ \frac{1}{e} - \frac{1}{e^2} & \text{if} \quad T > be^2. \end{cases}$$

The above competitive ratio is conservative by design: It was derived by taking the worst case over *unknown $\tau^*$* and the horizon length $T$ within each case. The proof of Theorem 3.2 in Appendix C.2 outlines the competitive ratio as a function of $\tau^*$ and $T$. Additionally, in Section 5, we investigate the impact of varying $\tau^*$ while keeping the horizon length fixed, providing a numerical illustration of how the expected competitive ratio changes. We note that the expected competitive ratio, averaged over the unknown $\tau^*$, is much better than our theoretical competitive ratio illustrated above. Based on our theoretical competitive ratio in

Theorem 3.2, we recommend choosing the horizon length $T$ relative to the budget $b$ to be below $be^2$, which aligns with our empirical findings in Section 5 (see Remark 5.1 for details).

*Remark 3.3.* As stated in Section 2.1, the term $\frac{1}{\tau^*}$ in the penalty can be replaced by a tunable strength parameter $\sigma$. In Section C.2, we show that for $T \leq be^2$, the above results hold over a wide range of $\sigma$ values, specifically $\sigma \leq \frac{b}{2}$. However, when $T > be^2$, $\sigma$ should be on the order of $\frac{1}{\tau^*}$, ensuring that the penalty term scales similarly to the budget term in the objective.

*Remark 3.4.* Establishing an upper bound on the performance of any randomized algorithm for the OUS problem is challenging due to the non-smooth nature of the objective function and the problem's three different operating regimes. In Appendix G, we derive a loose upper bound of $0.5$ for the OUS problem using Yao's lemma (Yao, 1977) and leave the derivation of a tighter bound for future work.

## 4. Learning-Augmented Algorithm

In this section, we propose a new approximation algorithm, Algorithm 2, under the learning-augmented setting, where we are provided with prediction confidence intervals $[L, U]$ for the unknown $\tau^*$. Algorithm 2 builds upon the non-learning augmented counterpart, Algorithm 1, utilizing the given confidence interval for optimization. Similar to Algorithm 1, we initialize $\alpha \sim [b, be]$ with density $1/\alpha$, and the current "guess" of $\tau^*$ is reflected by $\tilde{\tau} + L$.

In Algorithm 2, the three scenarios differ from those in Algorithm 1. Here, the distinction is based on the relationship between the upper bound of the interval, $U$, and the budget $b$. The three scenarios are 1) $U \leq be$ (**Subroutine 4**), 2) $be < U \leq be^2$, further divided into 2a) $U - L \leq b(e - 1)$ (**Subroutine 4**), and 2b) $U - L > b(e - 1)$ (**Subroutine 2**), and 3) $U > be^2$, further divided into 3a) $U - L \leq b(e + 1)$ (**Subroutine 5**), and 3b) $U - L > b(e + 1)$ (**Subroutine 6**).

Similarly, we first demonstrate that Algorithm 2 produces a feasible solution to Problem (1), with the proof provided in Appendix D.1 .

**Lemma 4.1.** *Let $p_i^{A2}$ be the probability returned by Algorithm 2 at risk time $i \in [\tau^*]$. This solution always satisfies*

**Algorithm 2** Randomized Online Algorithm With Prediction Confidence Intervals

1: **Input:** $T$, $b$, $[L, U]$
2: **Initialize:** $j = 1$, sample $\alpha \in [b, be]$ from a distribution with p.d.f. $f(\alpha) = 1/\alpha$, and initialize $\tilde{\tau} = \alpha$
3: **for** $i = 1, ..., \tau^*$ **do**
4:     We calculate:

$$\text{Int}(\tilde{\tau}) = \begin{cases} \lfloor \tilde{\tau} \rfloor & w.p. \quad \lceil \tilde{\tau} \rceil - \tilde{\tau} \\ \lceil \tilde{\tau} \rceil & w.p. \quad \tilde{\tau} - \lfloor \tilde{\tau} \rfloor \end{cases}$$

5:     **if** $U \leq be$ **then**
6:        Update $\tilde{\tau}$ and set $p_i$ using **Subroutine** 4
7:     **else if** $be < U \leq be^2$ **then**
8:        **if** $U - L \leq b(e - 1)$ **then**
9:           Update $\tilde{\tau}$ and set $p_i$ with **Subroutine** 4
10:        **else**
11:           Update $\tilde{\tau}$ and set $p_i$ with **Subroutine** 2
12:        **end if**
13:     **else**
14:        **if** $U - L \leq b(e + 1)$ **then**
15:           Update $\tilde{\tau}$ and set $p_i$ with **Subroutine** 5
16:        **else**
17:           Update $\tilde{\tau}, b$ and set $p_i$ with **Subroutine** 6
18:        **end if**
19:     **end if**
20:     Output sampling probability $p_i$
21: **end for**

---

**Subroutine 4** ($i, b, \tilde{\tau}, L, U, \text{Int}(\tilde{\tau})$)

1: **if** $i > \text{Int}(\tilde{\tau}) + L$ **then**
2:     $\tilde{\tau} = \tilde{\tau}e$
3: **end if**
4: $p_i = \frac{b}{\min(U, \tilde{\tau} + L)}$

---

**Subroutine 5** ($i, b, \tilde{\tau}, L, U, \text{Int}(\tilde{\tau})$)

1: **if** $i > \text{Int}(\tilde{\tau}) + L$ **then**
2:     $\tilde{\tau} = \tilde{\tau}e$
3: **end if**
4: $p_i = \frac{b}{\min(U, \tilde{\tau}e + L)}$

---

the budget constraint in expectation, i.e., $\mathbb{E}\left[\sum_{i=1}^{\tau^*} p_i^{A2}\right] \leq b$, where the expectation is taken over the randomness of the algorithm.

Next, we provide a theoretical guarantee on its performance:

**Theorem 4.2.** *Algorithm* 2 *is* 1-*consistent and* $\mathcal{X}(U)$-*robust, where* $\mathcal{X}(U)$ *is defined as follows:*

$$\mathcal{X}(U) := \begin{cases} \ln 2 + \frac{e-1}{e} \ln \frac{e-1}{e} & if \quad U \leq be, \\ \frac{1}{e} & if \quad be < U \leq be^2, \\ 2 - \ln(e^2 - e + 1) & if \quad U > be^2. \end{cases}$$

We first note that Algorithm 2 is 1-consistent, achieving the performance of the offline clairvoyant when the prediction is perfect. The proof of Theorem 4.2 in Appendix D.2 provides a detailed analysis of the competitive ratio, which depends on the parameters $\tau^*$, $L$, and $U$.[5] Furthermore, Section 5 explores the impact of varying the prediction confidence interval width $U - L$ while keeping $\tau^*$ constant. Our findings reveal that Algorithm 2 almost always outperforms Algorithm 1. Finally, we discuss the design choice

---

[5]In Theorem 4.3, we present the competitive ratios for scenarios 1), 2), and 3) separately, combining the results of the respective subroutines.

of $T$ relative to $b$ in the context of prediction intervals in Remark 5.2.

*Remark* 4.3. Similarly, the term $\frac{1}{\tau^*}$ in the penalty can be replaced by a tuning parameter $\sigma$. In Section D.2, we show that for $U \leq be^2$, the above results hold for a wide range of $\sigma$ values, specifically $\sigma \leq \frac{b}{e}$. However, when $T > be^2$, $\sigma$ should be of the order $\frac{1}{\tau^*}$ to align the penalty term with the budget term in the objective.

## 5. Experiments

In this section, we numerically assess the performance of our proposed algorithms through numerical experiments conducted on both synthetic and real-world datasets.

### 5.1. Synthetic Experiments

**Benchmarks** In the setting without learning augmentation, we compare Algorithm 1 against a conservative benchmark that delivers interventions with a constant probability $b/T$. In the learning-augmented setting, where a confidence interval $[L, U]$ is provided, we compare Algorithm 2 against two benchmarks: (1) a benchmark that delivers interventions with a constant probability $b/U$, and (2) Algorithm 1. Due to the limited algorithmic work on OUS (Online Uniformity Scheduling) and the absence of existing algorithms that handle confidence intervals, we do not include additional benchmarks in the synthetic data experiments. However, in the real-world example, we also evaluate the SeqRTS algorithm (Liao et al., 2018), which does not account for the prediction uncertainty of $\tau^*$. The metric used for the evaluation is the average competitive ratio.

**Without Learning Augmentation** In this setting, we evaluate the performance of Algorithm 1 across all three scenarios outlined in Theorem 3.2. To do this, we fix the budget at $b = 3$ and alter the horizon lengths $T$ to align with each scenario. For Scenarios 1 and 2, we set $T$ to the maximum allowable values with $b = 3$, specifically $T = 8$ and 22, as illustrated in Figure 1 (left and middle). For Scenario

**Subroutine 6** $(i, b, \tilde{\tau}, L, U, \text{Int}(\tilde{\tau}))$

1: **if** $i > \text{Int}(\tilde{\tau}) + L$ **then**
2:      $j = j + 1$
3:      **if** $j = 2$ **then**
4:          $b = b(1 - \frac{\tilde{\tau} + L - b}{\tilde{\tau}(e-1) + L})$
5:      **else**
6:          $b = b(1 - \frac{1}{e})$
7:      **end if**
8:      $\tilde{\tau} = \tilde{\tau} e$
9: **end if**
10: **if** $j = 1$ **then**
11:      $p_i = \frac{b}{\tilde{\tau}(e-1) + L}$
12: **else**
13:      $p_i = \frac{b}{\tilde{\tau} e}$
14: **end if**

3, where $T$ can grow asymptotically to infinity, we choose $T = 100$ for simplicity (Figure 1 right). To simulate risk occurrences, we randomly choose an integer $\tau^*$ from the interval $[b, T-1]$ and then select $\tau^*$ distinct time points uniformly at random from the $T$ available time steps as risk times.

Figure 1 displays the average competitive ratio across a range of $\tau^*$ values. Figure 1a indicates that our randomized algorithm consistently outperforms the benchmark by a constant competitive ratio for all values of $\tau^*$ in Scenario 1. Similarly, Figure 1b shows that in Scenario 2, our randomized algorithm increasingly outperforms the benchmark as $\tau^*$ deviates further from the horizon length $T$. In Figure 1c, as $T$ increases, the average competitive ratio of our algorithm remains constant and consistently outperforms the benchmark.[6] Therefore, we conclude that our algorithm increasingly outperforms the benchmark as $T$ grows to infinity.

*Remark* 5.1 (**Design choice of** $b$ **and** $T$ **in the absence of prediction confidence intervals**). In real-world applications, the intervention budget for each risk level is often fixed. However, a key design consideration is the choice of $T$, i.e., the granularity of the decision period. As illustrated in Figure 1, while Scenario 3 achieves the greatest performance improvement as $T$ approaches infinity, our randomized algorithm attains the highest competitive ratio across all $\tau^*$ in Scenarios 1 and 2. Thus, in the absence of prediction intervals, we recommend selecting $T$ such that $T \leq be^2$.

**With Learning Augmentation** In this setting, we evaluate the performance of Algorithms 1 and 2 across varying prediction interval widths. As in the non-learning-augmented setting, we fix the budget at $b = 3$ and ex-

---

[6]This is because when $b$ is fixed, the treatment assignment probability is independent of $T$.

amine the performance of our learning-augmented algorithm for $T = 8, 22$, and 100, covering the three scenarios outlined inAlgorithm 2. To compare the performance of our algorithm across various confidence widths, we fix $\tau^* = \text{Int}[0.5(T + b)]$ across all simulations.[7] The confidence intervals are randomly generated based on the given width and must contain $\tau^*$.

Figure 2 plots the average competitive ratio of each algorithm across a range of interval widths. We observe that the naive benchmark (where $p_i = b/U$ for all $i \in [\tau^*]$) outperforms the Algorithm 1 (which does not have access to the prediction interval) when the confidence interval is narrow. This is not surprising as in this case $\tau^* \approx U$. However, as the prediction interval widens, our Algorithm 1 outperforms the naive benchmark. In addition, we observe that our learning-augmented algorithm performs no worse than both the naive benchmark and the randomized algorithm. In particular, the advantage of Algorithm 2 is the largest in Scenario 3.

*Remark* 5.2 (**Design choice of** $b$ **and** $T$ **in presence of prediction intervals**). If we expect the value of $\tau^*$ to be small, we recommend setting $T \leq be^2$ to ensure that the algorithm always operates in Scenario 2, where $U \leq be^2$. If we expect a reasonably large value of $\tau^*$, we recommend setting a large value for $T > be^2$ such that the algorithm operates under Scenario 3, where $U$ can exceed $be^2$.

Additional experimental results for small $\tau^*$ are provided in Appendix E.1. We note that as $\tau^*$ decreases, the advantage of our algorithm in Scenario 2 increases. We also include competitive ratio figures without the penalization term from Problem (1) in Appendix E.2, measuring the fraction of the budget spent by our algorithms.

### 5.2. Real-World Experiments on HeartSteps

Our research is motivated by the Heartsteps V1 mobile health study, which aimed to increase physical activity among 37 sedentary individuals over a six-week period, with $T = 144$ decision points per day (Klasnja et al., 2019). At each decision time $t$, a risk variable $R_t$ is observed, which is binary: $R_t = 1$ indicates a sedentary state, identified by recording fewer than 150 steps in the prior 40 minutes, and $R_t = 0$ signifies a non-sedentary state. The total number of risk times, $\tau^* = \sum_{t=1}^{T} R_t$, is unknown. The primary objective here is to uniformly distribute approximately $b = 1.5$ interventions across sedentary times each day.

**Benchmarks** In addition to the naive benchmark $b/U$, we compare the performance of Algorithms 1 and 2 with the SeqRTS algorithm, as proposed by Liao et al. (2018). Under SeqRTS, the budget may be exhausted before all available

---

[7]If we allow $\tau^*$ to change across different simulations, then the difference that we observe in competitive ratio might be due to this change in $\tau^*$.

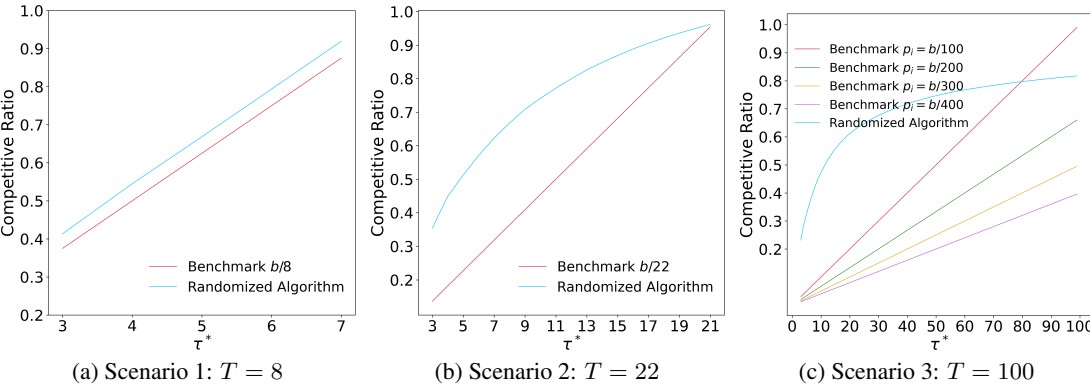

(a) Scenario 1: $T = 8$      (b) Scenario 2: $T = 22$      (c) Scenario 3: $T = 100$

*Figure 1.* Average competitive ratio under non-learning augmented setting with $b = 3$. The scenarios correspond to $T \leq be$, $be < T \leq be^2$, and $T > be^2$, respectively.

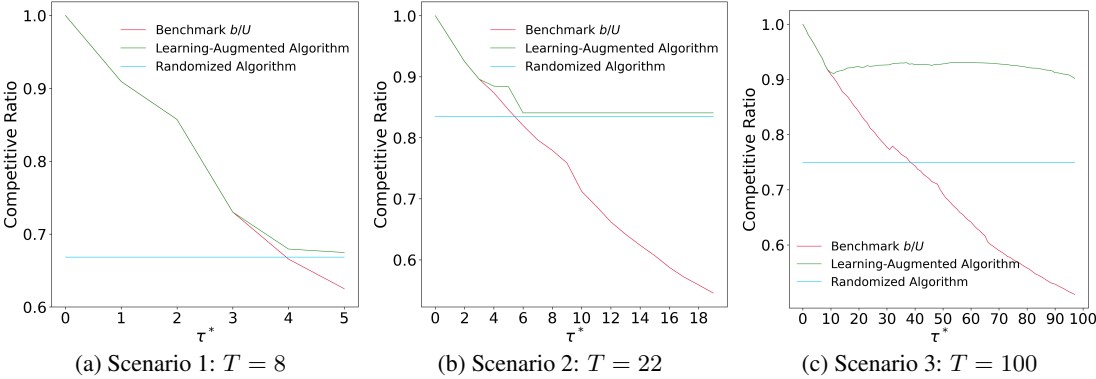

(a) Scenario 1: $T = 8$      (b) Scenario 2: $T = 22$      (c) Scenario 3: $T = 100$

*Figure 2.* Average competitive ratio under learning augmented setting with $b = 3$. The scenarios correspond to $U \leq be$, $be < U \leq be^2$, and $U > be^2$, respectively.

risk times are allocated. In such cases, a minimum probability of $1 \times 10^{-6}$ is assigned to the remaining risk times when evaluating the objective in Problem (1). A comprehensive description of the SeqRTS method and additional implementation details are provided in Appendix F. Performance is assessed using the competitive ratio and the average entropy change across user days.

In Figure 3, Algorithm 2, which incorporates a prediction interval, invariably outperforms the non-learning counterpart, the SeqRTS approach, and the naive benchmark $b/U$. Moreover, our proposed algorithms exhibit superior uniformity in risk times sampling, evidenced by reduced entropy change compared to both the non-learning algorithm and SeqRTS, as further detailed in Figure 7 in Appendix F. To better understand the behavior of SeqRTS, we set the minimum probability to 0 in Figure 8 in Section F. This figure illustrates that SeqRTS could deplete its budget even when the prediction is fairly accurate, highlighting the robustness of our algorithms under adversarial risk level arrivals.

**Conclusion and Future Works** This paper marks the first attempt to study the online uniform allocation problem within the framework of approximation algorithms. We in-

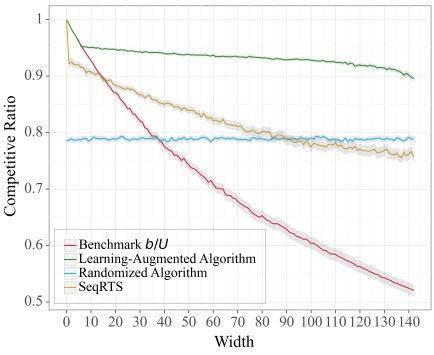

*Figure 3.* Average competitive ratio across user days under various prediction interval widths on HeartSteps V1 dataset. The shaded area indicates the $\pm 1.96$ standard error bounds across user days.

troduce two novel online algorithms—either incorporating learning augmentation or not—backed by rigorous theoretical guarantees and empirical results. Future works include adapting existing algorithms to scenarios where prediction intervals improve over time.

## Impact Statement

This paper presents work whose goal is to advance the field of Machine Learning. There are many potential societal consequences of our work, none which we feel must be specifically highlighted here.

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

## A. Extension to Multiple Risk Levels

In this section, we discuss the extension of the online uniform risk times sampling problem to multiple risk levels.

At each time $t \in [1, T]$, the patient is associated with an ordinal risk level from $K$ possible levels. The higher the risk level, the more likely the patient will experience a negative event, such as a relapse to smoking. As stated previously, the distributions of risk levels are allowed to change *arbitrarily* across decision periods since we anticipate that the treatment will reduce subsequent risk.

Let $\tau_k^*$ be the *unknown true* number of decision times at risk level $k \in [K]$ in a decision period, which is revealed at the end of the horizon $T$. For each risk level $k$, we define $p_{k,i_k} \in (0,1)$ to be the treatment probability at time $i_k \in [\tau_k^*]$. The algorithm is provided with a *soft* budget of $b_k$ for each risk level $k$, representing the total *expected* number of interventions allowed to be delivered at risk level $k$ within each decision period. As before, we assume $\tau_k^* > b_k$ for technical convenience (Liao et al., 2018).

Then at each decision time $i_k$, the algorithm decides the intervention probability $p_{k,i_k}$. For each risk level $k$, the objectives of the online uniform allocation problem are to 1) assign the intervention probabilities $\{p_{k,i_k}\}_{i_k \in [\tau_k^*]}$ as uniform as possible across risk times, and 2) maximize the sum of intervention probabilities across risk times while adhering to the budget constraint $b_k$.

For every risk level $k \in [K]$, we define the following optimization problem:

$$\max \sum_{i_k}^{\tau_k^*} p_{k,i_k} - \frac{1}{\tau_k^*} \ln \left( \frac{\max_{i_k \in [\tau_k^*]} p_{k,i_k}}{\min_{i_k \in [\tau_k^*]} p_{k,i_k}} \right)$$

$$\text{s.t. } \mathbb{E}\left[ \sum_{i_k=1}^{\tau_k^*} p_{k,i_k} \right] \leq b_k$$

$$p_{k,i_k} \in (0,1) \quad \forall i \in [\tau_k^*]. \tag{4}$$

Notably, the proposed algorithms offer a feasible solution to the above optimization problem, allowing us to address each risk level independently.

## B. The Penalty Term for Uniformity

We have previously considered statistical distance measures for quantifying the uniformity objective. One important measure is the Kullback-Leibler (KL) divergence. However, this measure is not well defined in our setting since the optimal solution (which is a point mass on $b/\tau^*$) and the solutions given by our proposed algorithms are not defined on the same sample space.

Recall that for two discrete distributions $P$ and $Q$ defined on the same sample space $\mathcal{X}$, the KL divergence is given by

$$D_{KL}(P\|Q) = \sum_{x \in \mathcal{X}} P(x) \log \frac{P(x)}{Q(x)},$$

where $P$ represents the data distribution, i.e., the optimal solution, and $Q$ represents an approximation of $P$, i.e., the solution given by an algorithm.

Let us consider a toy example where $\tau^* = b(e-1)$. In this case, the optimal solution should be $p_i = \frac{b}{b(e-1)} = \frac{1}{e-1}$ for each risk time $i \in [\tau^*]$. The corresponding distribution is a point mass, meaning the sample space $\mathcal{X}$ consists of a single element $(p_1 = \frac{1}{e-1}, \cdots, p_{\tau^*} = \frac{1}{e-1})$ with probability 1. The solutions given by our proposed algorithms are of the form $(p_1, \cdots, p_{\tau^*})$, but the sample space $\mathcal{X}$ is $Q^{\tau^*}$, where the support of $Q$ is $(0,1)$.

Clearly, the optimal solution and the solutions given by the proposed algorithms are not defined on the same sample space. Therefore, the KL divergence is not well-defined in this context.

## C. Proof for Algorithm 1

### C.1. Proof of Lemma 3.1: Budget constraint

*Proof.* We prove that the budget constraint is satisfied in expectation under each subroutine in Algorithm 1.

**Subroutine 1**

Recall that $\tau^*$ is the true number of risk times. Here, we suppose $\tau^* = \beta e^{j^*}$ for some $j^* \in \mathbb{Z}^+$ and $\beta \in [b, be]$. Since $T \leq be$, we have that $j^* = 0$.

In this analysis, our focus is solely on the worst-case scenario, where both $T$ and $\tau^*$ are very close to $be$. Assuming $\eta > b$ (where $\eta = \frac{T}{e-1}$), if this is not the case, the algorithm uniformly sets $p_i = \frac{b}{T}$ throughout.

When $\alpha$ falls in the range of $[b, \eta]$, Algorithm 1 starts with $p_i = \frac{b}{\alpha(e-1)}$ with a running length of $\alpha$, then transitions to $p_i = \frac{b}{T}$ for the second phase with a running length of $\beta - \alpha$. Otherwise, it consistently assigns $p_i = \frac{b}{T}$ with a running length of $\beta$. Therefore, the expected budget consists of two parts: one for $\alpha \in [b, \eta]$, where the term inside represents the used budget or the sum of treatment probabilities, and the other for $\alpha \in [\eta, be]$, where the term inside represents the sum of treatment probabilities:

$$
\begin{aligned}
\mathbb{E}[\text{Budget}] &= \int_{\eta}^{be} \frac{b}{T}\beta\frac{1}{\alpha}d\alpha + \int_{b}^{\eta} \left[\frac{b}{\alpha(e-1)}\alpha + \frac{b}{T}(\beta - \alpha)\right]\frac{1}{\alpha}d\alpha \\
&= \frac{b\beta}{T}\ln\frac{be}{\eta} + \frac{b}{e-1}\ln\frac{\eta}{b} + \frac{b\beta}{T}\ln\frac{\eta}{b} - \frac{b}{T}(\eta - b) \\
&= \frac{b\beta}{T} + \frac{b}{e-1}\ln\frac{\eta}{b} - \frac{b\eta}{T} + \frac{b^2}{T} \\
&\leq b + \frac{b}{e-1}\ln\frac{T}{b(e-1)} - \frac{b}{e-1} + \frac{b^2}{T} \quad \text{increasing with } \beta \ (\beta = T) \\
&\leq b - \frac{b}{e-1} + \frac{b}{e-1} - \frac{b}{e-1}\ln(e-1) + \frac{b}{e} \quad \text{increasing with } T \ (T = be) \\
&\leq b - \frac{b}{e-1}\ln(e-1) + \frac{b}{e} \\
&\approx b.
\end{aligned}
$$

**Subroutine 2**

Reiterating our initial assumption, we set $\tau^* = \beta e^{j^*}$. The condition $be < T \leq be^2$ limits $j^*$ to either 0 or 1. However, our analysis is particularly concerned with the worst-case scenario, hence we consider only the case where $j^* = 1$.

When $\alpha$ falls in the range of $[b, \beta]$, Algorithm 1 starts with $p_i = \frac{b}{\alpha(e-1)}$ with a running length of $\alpha$, transitions to $p_i = \frac{b}{\alpha e(e-1)}$ with a running length of $\alpha e - \alpha$, and then continues with $p_i = \frac{b}{e^3}$ with a running length of $\beta e - \alpha e$. Otherwise, Algorithm 1 starts with $p_i = \frac{b}{\alpha(e-1)}$ with a running length of $\alpha$, transitions to $p_i = \frac{b}{\alpha e(e-1)}$ with a running length of $\beta e - \alpha$.

The expected budget is therefore

$$\mathbb{E}\left[\text{Budget}\right] = \int_{\beta}^{be} \left[ \frac{b}{\alpha(e-1)}\alpha + \frac{b}{\alpha e(e-1)}(\beta e - \alpha) \right] \frac{1}{\alpha} d\alpha + \int_{b}^{\beta} \left[ \frac{b}{\alpha(e-1)}\alpha \right.$$

$$\left. + \frac{b}{\alpha e(e-1)}(\alpha e - \alpha) + \frac{b}{\alpha e^3}(\beta e - \alpha e) \right] \frac{1}{\alpha} d\alpha$$

$$= \frac{b}{e-1} + \frac{b\beta}{e-1}(\frac{1}{\beta} - \frac{1}{be}) - \frac{b}{e(e-1)}\ln\frac{be}{\beta} + \frac{b}{e}\ln\frac{\beta}{b} + \frac{b\beta}{e^2}(\frac{1}{b} - \frac{1}{\beta}) - \frac{b}{e^2}\ln\frac{\beta}{b}$$

$$= \frac{b}{e-1} + \frac{b}{e-1} - \frac{\beta}{e(e-1)} + \frac{\beta}{e^2} - \frac{b}{e^2} - \frac{b}{e(e-1)} + (\frac{b}{e} + \frac{b}{e(e-1)} - \frac{b}{e^2})\ln\frac{\beta}{b}$$

$$= \frac{b}{e-1} + \frac{b}{e} - \frac{b}{e^2} - \frac{\beta}{e^2(e-1)} + (\frac{b}{e-1} - \frac{b}{e^2})\ln\frac{\beta}{b}$$

$$\leq \frac{b}{e-1} + \frac{b}{e} - \frac{b}{e^2} - \frac{be}{e^2(e-1)} + (\frac{b}{e-1} - \frac{b}{e^2})\ln\frac{be}{b} \quad \text{increasing with } \beta \ (\beta = be)$$

$$= \frac{2b}{e} + \frac{b}{e-1} - \frac{2b}{e^2} \approx b$$

**Subroutine 3**

Under the assumption of $\tau^* = \beta e^{j^*}$, and given the condition $T > be^2$, it is possible for $j^*$ to be 0 or to extend towards infinity. Our focus, however, is confined to the worst-case scenarios, particularly those where $j^* \geq 1$.

When $\alpha$ falls in the range of $[b, \beta]$. Algorithm 1 stops with $p_i = \frac{b(1-1/e)^{j^*}}{\alpha e^{j^*+2}}$ with a running length of $\beta e^{j^*} - \alpha e^{j^*}$. Otherwise, Algorithm 1 stops with $p_i = \frac{b(1-1/e)^{j^*-1}}{\alpha e^{j^*+1}}$ with a running length of $\beta e^{j^*} - \alpha e^{j^*-1}$. Therefore, the expected budget is

$$\mathbb{E}\left[\text{Budget}\right] = \int_{\beta}^{be} \left[ \frac{b}{\alpha e}\alpha + \sum_{i=2}^{j^*} \frac{b(1-1/e)^{j-2}}{\alpha e^j}(\alpha e^{j-1} - \alpha e^{j-2}) + \frac{b(1-1/e)^{j^*-1}}{\alpha e^{j^*+1}}(\beta e^{j^*} - \alpha e^{j^*-1}) \right] \frac{1}{\alpha} d\alpha$$

$$+ \int_{b}^{\beta} \left[ \frac{b}{\alpha e}\alpha + \sum_{j=2}^{j^*+1} \frac{b(1-1/e)^{j-2}}{\alpha e^j}(\alpha e^{j-1} - \alpha e^{j-2}) + \frac{b(1-1/e)^{j^*}}{\alpha e^{j^*+2}}(\beta e^{j^*} - \alpha e^{j^*}) \right] \frac{1}{\alpha} d\alpha$$

$$= \int_{\beta}^{be} \left[ \frac{b}{e} + b(1 - \frac{1}{e} - (1 - \frac{1}{e})^{j^*}) + \frac{b(e-1)^{j^*-1}}{e^{j^*+1}}\frac{\beta e - \alpha}{\alpha} \right] \frac{1}{\alpha} d\alpha$$

$$+ \int_{b}^{\beta} \left[ \frac{b}{e} + b(1 - \frac{1}{e} - (1 - \frac{1}{e})^{j^*+1}) + \frac{b(e-1)^{j^*}}{e^{j^*+2}}\frac{\beta - \alpha}{\alpha} \right] \frac{1}{\alpha} d\alpha$$

$$= \frac{b}{e} + b(1 - \frac{1}{e}) - b(1 - \frac{1}{e})^{j^*}\ln\frac{be}{\beta} - b(1 - \frac{1}{e})^{j^*+1}\ln\frac{\beta}{b}$$

$$+ \frac{b(e-1)^{j^*-1}}{e^{j^*+1}}\left[ e - \frac{\beta}{b} + \ln\frac{\beta}{be} \right] + \frac{b(e-1)^{j^*}}{e^{j^*+2}}\left[ \frac{\beta}{b} - 1 + \ln\frac{b}{\beta} \right]$$

$$\leq b - b(1 - \frac{1}{e})^{j^*+1}\ln\frac{be}{b} + \frac{b(e-1)^{j^*}}{e^{j^*+2}}\left[ \frac{be}{b} - 1 + \ln\frac{b}{be} \right] \quad \text{increasing with } \beta \ (\beta = be)$$

$$= b - b(1 - \frac{1}{e})^{j^*+1} + \frac{b(e-1)^{j^*}(e-2)}{e^{j^*+2}} \leq b$$

By combining the above results, we establish Lemma 3.1. □

**C.2. Proof of Theorem 3.1: Competitive Ratio**

*Proof.* In what follows we derive the competitive ratio under each subroutine.

**Subroutine 1**

Recall that $\tau^*$ represents the true number of available risk times at risk level $k$, we assume $\tau^* = \beta e^{j^*}$, where $j^* \in \mathbb{Z}^+$ and $\beta \in [b, be]$. It's evident that when $T \leq be$, $j^* = 0$ follows naturally.

Define $\eta = T/(e-1)$. Let us first consider the case where $\eta \leq b$, leading to $T \leq b(e-1)$. Given that $p_i = \frac{b}{\min(T, \hat{\tau}(e-1))}$, it follows that the algorithm consistently sets $p_i = \frac{b}{T}$. Consequently, we have

$$\mathbb{E}[\text{SOL}] = \frac{b}{T}\beta \geq \frac{b}{b(e-1)}\beta \geq \frac{b}{e-1}.$$

Next, let us consider the case where $\eta > b$. We focus on two cases: (1) $\beta < \eta$ and (2) $\beta \geq \eta$.

Suppose $\beta < \eta$. When $\alpha$ falls within $[b, \beta]$, the algorithm initiates with $p_i = \frac{b}{\alpha(e-1)}$ with a running length of $\alpha$, then adjusts to $p_i = \frac{b}{T}$ in the subsequent round with a running length of $\beta - \alpha$; when $\alpha$ falls in the range of $[\beta, \eta]$, the algorithm initiates with $p_i = \frac{b}{\alpha(e-1)}$ with a running length of $\beta$ and stops on this stage; otherwise, it consistently uses $p_i = \frac{b}{T}$ with a running length of $\beta$. Therefore, the expected solution is

$$\begin{aligned}
\mathbb{E}[\text{SOL}] &= \int_\eta^{be} \frac{b}{T}\beta\frac{1}{\alpha}d\alpha + \int_b^\beta \left[\frac{b}{\alpha(e-1)}\alpha + \frac{b}{T}(\beta-\alpha) - \sigma\ln\frac{T}{\alpha(e-1)}\right]\frac{1}{\alpha}d\alpha \\
&\quad + \int_\beta^\eta \frac{b}{\alpha(e-1)}\beta\frac{1}{\alpha}d\alpha \\
&= \frac{b\beta}{T}\ln\frac{be}{\eta} + \frac{b}{e-1}\ln\frac{\beta}{b} + \frac{b\beta}{T}\ln\frac{\beta}{b} - \frac{b}{T}(\beta-b) + \frac{\sigma}{2}\left(\ln(\frac{T}{\beta(e-1)})^2 - \ln(\frac{T}{b(e-1)})^2\right) \\
&\quad + \frac{b\beta}{e-1}(\frac{1}{\beta} - \frac{1}{\eta}) \\
&\geq \frac{b^2}{T}\ln\frac{be(e-1)}{T} + \frac{b}{e-1} - \frac{b^2}{T} \quad \text{increasing with } \beta \ (\beta = b) \\
&\geq \frac{b}{e}\ln(e-1) + \frac{b}{e(e-1)} \quad \text{decreasing with } T \ (T = be).
\end{aligned}$$

Suppose $\beta \geq \eta$. It follows that the algorithm always proceeds to the second round. When $\alpha$ falls within $[b, \eta]$, the algorithm initiates with $p_i = \frac{b}{\alpha(e-1)}$ with a running length of $\alpha$, then adjusts to $p_i = \frac{b}{T}$ in the subsequent round with a running length of $\beta - \alpha$; otherwise, it consistently uses $p_i = \frac{b}{T}$ with a running length of $\beta$. Consequently, we have

$$\begin{aligned}
\mathbb{E}[\text{SOL}] &= \int_\eta^{be} \frac{b}{T}\beta\frac{1}{\alpha}d\alpha + \int_b^\eta \left[\frac{b}{\alpha(e-1)}\alpha + \frac{b}{T}(\beta-\alpha) - \sigma\ln\frac{T}{\alpha(e-1)}\right]\frac{1}{\alpha}d\alpha \\
&= \frac{b\beta}{T}\ln\frac{be}{\eta} + \frac{b}{e-1}\ln\frac{\eta}{b} + \frac{b\beta}{T}\ln\frac{\eta}{b} - \frac{b}{T}(\eta-b) + \frac{\sigma}{2}\left(\ln(\frac{T}{\eta(e-1)})^2 - \ln(\frac{T}{b(e-1)})^2\right) \\
&\geq \frac{b}{e-1}\ln\frac{be}{\eta} + \frac{2b}{e-1}\ln\frac{T}{b(e-1)} - \frac{b}{e-1} + \frac{b^2}{T} - \frac{\sigma}{2}\ln(\frac{T}{b(e-1)})^2 \\
&\quad \text{increasing with } \beta \ (\beta = \eta = T/(e-1)) \\
&\geq \frac{2b}{e-1} - \frac{b}{e-1}\ln(e-1) - \frac{b}{e(e-1)} - \frac{\sigma}{2}\ln(\frac{e}{e-1})^2 \quad \text{decreasing with } T \ (T = be).
\end{aligned}$$

**Subroutine 2**

Recall that for $be < T \leq be^2$, $j^*$ is restricted to being either $0$ or $1$. Below we separately consider these two cases.

Suppose $j^* = 0$. When $\alpha$ falls in the range of $[b, \beta]$, Algorithm 1 begins with $p_i = \frac{b}{\alpha(e-1)}$ with a running length of $\alpha$, then transitions to $p_i = \frac{b}{\alpha e(e-1)}$ with a running length of $\beta - \alpha$. Otherwise, Algorithm 1 begins with $p_i = \frac{b}{\alpha(e-1)}$ with a running

length of $\beta$ and stops. It follows that

$$\mathbb{E}[\text{SOL}] = \int_{\beta}^{be} \frac{b}{\alpha(e-1)} \beta \frac{1}{\alpha} d\alpha + \int_{b}^{\beta} \left[ \frac{b}{\alpha(e-1)} \alpha + \frac{b}{\alpha e(e-1)} (\beta - \alpha) - \sigma \ln(e) \right] \frac{1}{\alpha} d\alpha$$

$$= \frac{b\beta}{e-1} \left( \frac{1}{\beta} - \frac{1}{be} \right) + \frac{b}{e-1} (\ln \beta - \ln b) + \frac{b\beta}{e(e-1)} \left( \frac{1}{b} - \frac{1}{\beta} \right)$$

$$- \frac{b}{e(e-1)} (\ln \beta - \ln b) - \sigma \ln \frac{\beta}{b}$$

$$= \frac{b}{e} + \frac{b}{e} \ln \frac{\beta}{b} - \sigma \ln \frac{\beta}{b}$$

$$\geq \frac{b}{e} \quad \text{increasing with } \beta \ (\beta = b).$$

Suppose $j^* = 1$. When $\alpha$ falls in the range of $[b, \beta]$, Algorithm 1 begins with $p_i = \frac{b}{\alpha(e-1)}$ with a running length of $\alpha$, transitions to $p_i = \frac{b}{\alpha e(e-1)}$ with a running length of $\alpha e - \alpha$, and then continues with $p_i = \frac{b}{e^3}$ with a running length of $\beta e - \alpha e$. Otherwise, Algorithm 1 begins with $p_i = \frac{b}{\alpha(e-1)}$ with a running length of $\alpha$, then transitions to $p_i = \frac{b}{\alpha e(e-1)}$ with a running length of $\beta e - \alpha$. Therefore, the expected solution is

$$\mathbb{E}[\text{SOL}] = \int_{\beta}^{be} \left[ \frac{b}{\alpha(e-1)} \alpha + \frac{b}{\alpha e(e-1)} (\beta e - \alpha) - \sigma \ln e \right] \frac{1}{\alpha} d\alpha$$

$$+ \int_{b}^{\beta} \left[ \frac{b}{\alpha(e-1)} \alpha + \frac{b}{\alpha e(e-1)} (\alpha e - \alpha) + \frac{b}{\alpha e^3} (\beta e - \alpha e) - \sigma \ln \frac{e^3}{e-1} \right] \frac{1}{\alpha} d\alpha$$

$$= \frac{b}{e-1} \ln \frac{be}{\beta} + \frac{b\beta}{e-1} \left( \frac{1}{\beta} - \frac{1}{be} \right) - \frac{b}{e(e-1)} \ln \frac{be}{\beta} - \sigma \ln \frac{be}{\beta}$$

$$+ \frac{b}{e-1} \ln \frac{\beta}{b} + \frac{b}{e} \ln \frac{\beta}{b} + \frac{b\beta}{e^2} \left( \frac{1}{b} - \frac{1}{\beta} \right) - \frac{b}{e^2} \ln \frac{\beta}{b} - \sigma \ln \frac{e^3}{e-1} \ln \frac{\beta}{b}$$

$$= \frac{b}{e} \ln \frac{be}{\beta} + \frac{b}{e-1} - \frac{\beta}{e(e-1)} - \sigma \ln \frac{be}{\beta}$$

$$+ \frac{b}{e-1} \ln \frac{\beta}{b} + \frac{b}{e} \ln \frac{\beta}{b} + \frac{\beta}{e^2} - \frac{b}{e^2} - \frac{b}{e^2} \ln \frac{\beta}{b} - \sigma \ln \frac{e^3}{e-1} \ln \frac{\beta}{b}$$

$$\geq \frac{b}{e} + \frac{b}{e-1} - \frac{b}{e(e-1)} - \sigma \quad \text{increasing with } \beta \ (\beta = b)$$

$$= \frac{2b}{e} - \sigma.$$

**Subroutine 3**

In the scenarios where $T > be^2$, we consider two cases: (1) $j^* \geq 1$ and (2) $j^* = 0$.

Let us first consider the case where $j^* \geq 1$. If $\alpha \geq \beta$, the algorithm stops at the $j^* + 1$th round by design of the algorithm ($\alpha e^{j^*} \geq \beta e^{j^*}$); on the other hand, if $\alpha < \beta$, the algorithm stops at the $j^* + 2$th round ($\alpha e^{j^*+1} \geq \beta e^{j^*}$). The objective function when $\alpha \geq \beta$ is

$$\text{SOL}_1 = \sum_{j=1}^{j^*} \frac{b \left(1 - \frac{1}{e}\right)^{j-2}}{\alpha e^j} \left( \alpha e^{j-1} - \alpha e^{j-2} \right) + \frac{b \left(1 - \frac{1}{e}\right)^{j^*-1}}{\alpha e^{j^*+1}} \left( \beta e^{j^*} - \alpha e^{j^*-1} \right) - \sigma \ln \frac{e^{2j^*-1}}{(e-1)^{j^*-1}}$$

$$= \sum_{j=1}^{j^*} \frac{b(e-1)^{j-1}}{e^j} + \frac{b(e-1)^{j^*-1}}{e^{j^*+1}} \frac{\beta e - \alpha}{\alpha} - \sigma \ln \frac{e^{2j^*-1}}{(e-1)^{j^*-1}}$$

$$= b \left( 1 - \left(1 - \frac{1}{e}\right)^{j^*} \right) + \frac{b(e-1)^{j^*-1}}{e^{j^*+1}} \frac{\beta e - \alpha}{\alpha} - \sigma \ln \frac{e^{2j^*-1}}{(e-1)^{j^*-1}}.$$

The objective function when $\alpha < \beta$ is

$$
\begin{aligned}
\text{SOL}_2 &= \sum_{j=1}^{j^*+1} \frac{b\left(1-\frac{1}{e}\right)^{j-2}}{\alpha e^j}\left(\alpha e^{j-1} - \alpha e^{j-2}\right) + \frac{b\left(1-\frac{1}{e}\right)^{j^*}}{\alpha e^{j^*+2}}\left(\beta e^{j*} - \alpha e^{j*}\right) - \sigma \ln \frac{e^{2j^*+1}}{(e-1)^{j^*}} \\
&= \sum_{j=1}^{j^*+1} \frac{b(e-1)^{j-1}}{e^j} + \frac{b(e-1)^{j*}}{e^{j^*+2}} \frac{\beta-\alpha}{\alpha} - \sigma \ln \frac{e^{2j^*+1}}{(e-1)^{j^*}} \\
&= b\left(1 - \left(1-\frac{1}{e}\right)^{j^*+1}\right) + \frac{b(e-1)^{j*}}{e^{j^*+2}} \frac{\beta-\alpha}{\alpha} - \sigma \ln \frac{e^{2j^*+1}}{(e-1)^{j^*}}.
\end{aligned}
$$

The expected value of our solution is

$$
\mathbb{E}[\text{SOL}] = \int_\beta^{be} \text{SOL}_1\, f(\alpha)d\alpha + \int_b^\beta \text{SOL}_2\, f(\alpha)d\alpha. \tag{5}
$$

Notice that

$$
\begin{aligned}
\int_\beta^{be} \text{SOL}_1\, f(\alpha)d\alpha &= b\left(1 - \left(1-\frac{1}{e}\right)^{j^*}\right)\int_\beta^{be} \frac{1}{\alpha}d\alpha + \frac{b(e-1)^{j^*-1}}{e^{j^*+1}} \int_\beta^{be} \frac{\beta e - \alpha}{\alpha} \frac{1}{\alpha}d\alpha \\
&\quad - \left[\sigma \ln \frac{e^{2j^*-1}}{(e-1)^{j^*-1}}\right] \int_\beta^{be} \frac{1}{\alpha}d\alpha \\
&= b\left(1 - \left(1-\frac{1}{e}\right)^{j^*}\right) \ln \frac{be}{\beta} + \frac{b(e-1)^{j^*-1}}{e^{j^*+1}}\left(e - \frac{\beta}{b} - \ln \frac{be}{\beta}\right) \\
&\quad - \left[\sigma \ln \frac{e^{2j^*-1}}{(e-1)^{j^*-1}}\right] \ln \frac{be}{\beta}.
\end{aligned}
$$

and

$$
\begin{aligned}
\int_b^\beta \text{SOL}_2\, f(\alpha)d\alpha &= b\left(1 - \left(1-\frac{1}{e}\right)^{j^*+1}\right)\int_b^\beta \frac{1}{\alpha}d\alpha + \frac{b(e-1)^{j^*}}{e^{j^*+2}} \int_b^\beta \frac{\beta-\alpha}{\alpha} \frac{1}{\alpha}d\alpha \\
&\quad - \left[\sigma \ln \frac{e^{2j^*+1}}{(e-1)^{j^*}}\right] \int_b^\beta \frac{1}{\alpha}d\alpha \\
&= b\left(1 - \left(1-\frac{1}{e}\right)^{j^*+1}\right) \ln \frac{\beta}{b} + \frac{b(e-1)^{j^*}}{e^{j*+2}}\left(\frac{\beta}{b} - 1 - \ln \frac{\beta}{b}\right) \\
&\quad - \left[\sigma \ln \frac{e^{2j^*+1}}{(e-1)^{j^*}}\right] \ln \frac{\beta}{b}.
\end{aligned}
$$

Hence,

$$\mathbb{E}[\text{SOL}] = b\left(1 - \left(1 - \frac{1}{e}\right)^{j^*}\right)\ln\frac{be}{\beta} + b\left(1 - \left(1 - \frac{1}{e}\right)^{j^*+1}\right)\ln\frac{\beta}{b}$$

$$+ \frac{b(e-1)^{j^*-1}}{e^{j^*+1}}\left(e - \frac{\beta}{b} - \ln\frac{be}{\beta}\right) + \frac{b(e-1)^{j^*}}{e^{j^*+2}}\left(\frac{\beta}{b} - 1 - \ln\frac{\beta}{b}\right)$$

$$- \left[\sigma\ln\frac{e^{2j^*-1}}{(e-1)^{j^*-1}}\right]\ln\frac{be}{\beta} - \left[\sigma\ln\frac{e^{2j^*+1}}{(e-1)^{j^*}}\right]\ln\frac{\beta}{b}$$

$$= b - b\left(1 - \frac{1}{e}\right)^{j^*}\left[\ln\frac{be}{\beta} + \left(1 - \frac{1}{e}\right)\ln\frac{\beta}{b}\right]$$

$$+ \frac{b(e-1)^{j^*-1}}{e^{j^*+1}}\left[e - \frac{\beta}{b} - \ln\frac{be}{\beta} + \frac{e-1}{e}\left(\frac{\beta}{b} - 1 - \ln\frac{\beta}{b}\right)\right]$$

$$- \left[\sigma\ln\frac{e^{2j^*-1}}{(e-1)^{j^*-1}}\right]\ln\frac{be}{\beta} - \left[\sigma\ln\frac{e^{2j^*+1}}{(e-1)^{j^*}}\right]\ln\frac{\beta}{b}$$

$$\geq b - b\left(1 - \frac{1}{e}\right)^{j^*} + b\left(1 - \frac{1}{e}\right)^{j^*}\frac{e-2}{e(e-1)} - \sigma\ln\frac{e^{2j^*-1}}{(e-1)^{j^*-1}} \quad \text{increasing with } \beta \ (\beta = b).$$

Now consider the case where $j^* = 0$. When $\alpha$ falls within $[b, \beta]$, Algorithm 1 starts with $p_i = \frac{b}{\alpha e}$ with a running length of $\alpha$, then transitions to $p_i = \frac{b}{\alpha e^2}$ with a running length of $\beta - \alpha$. Otherwise, Algorithm 1 keeps $p_i = \frac{b}{\alpha e}$ for $\beta$ time points. It follows that

$$\mathbb{E}[\text{SOL}] = \int_\beta^{be} \frac{b}{\alpha e}\beta\frac{1}{\alpha}d\alpha + \int_b^\beta \left[\frac{b}{\alpha e}\alpha + \frac{b}{\alpha e^2}(\beta - \alpha) - \sigma\ln e\right]\frac{1}{\alpha}d\alpha$$

$$= \frac{b\beta}{e}\left(\frac{1}{\beta} - \frac{1}{be}\right) + \frac{b}{e}\ln\frac{\beta}{b} + \frac{b\beta}{e^2}\left(\frac{1}{b} - \frac{1}{\beta}\right) - \frac{b}{e^2}\ln\frac{\beta}{b} - \sigma\ln\frac{\beta}{b}$$

$$= \frac{b}{e} - \frac{\beta}{e^2} + \frac{\beta}{e^2} - \frac{b}{e^2} + \left(\frac{b}{e} - \frac{b}{e^2}\right)\ln\frac{\beta}{b} - \sigma\ln\frac{\beta}{b}$$

$$\geq b\left(\frac{1}{e} - \frac{1}{e^2}\right) \quad \text{increasing with } \beta \ (\beta = b).$$

**Tuning parameter selection** For Scenario 1) where $T \leq be$, the competitive ratio is the

$$\min\left(\frac{1}{e}\left(\ln(e-1) + \frac{1}{e-1}\right), \frac{2}{e-1} - \frac{1}{e-1}\ln(e-1) - \frac{1}{e(e-1)} - \frac{\sigma}{b}(1 - \ln(e-1))\right).$$

For Scenario 2) where $be < T \leq be^2$, the competitive ratio is

$$\min\left(\frac{1}{e}, \frac{2}{e} - \frac{\sigma}{b}\right).$$

For Scenario 3) where $T > be^2$, the competitive ratio is

$$\min\left(\frac{1}{e} - \frac{1}{e^2}, 1 - (1 - \frac{1}{e})^{j^*} + (1 - \frac{1}{e})^{j^*}\frac{e-2}{e(e-1)} - \frac{\sigma}{b}\ln\frac{e^{2j^*-1}}{(e-1)^{j^*-1}}\right).$$

By restricting the value of $\sigma$ under each scenario and combining the above results, we establish Theorem 3.2. Specifically, when $\sigma = \frac{1}{\tau^*}$, it can be verified that Theorem 3.2 holds. $\qquad\square$

## D. Proof for Algorithm 2

### D.1. Proof of Lemma 4.1: Budget constraint

*Proof.* We prove that the budget constraint is satisfied in expectation under each subroutine in Algorithm 2.

**Subroutine 4** Let us suppose that $\tau = L + \beta e^{j^*}$ for some $j^* \in \mathbb{Z}^+$ and $\beta \in [b, be]$. Note that this implicitly implies that $\tau \geq L + b$, as we only consider the worst case where $\tau^*$ is large enough. Define $\delta = U - L$. Under the condition $U \leq be$ or $\delta \leq b(e-1)$, we have $j^* = 0$.

When $\delta \leq b$, Algorithm 2 would consistently use $p_i = \frac{b}{U}$, and the budget constraint is satisfied obviously. Now suppose $\delta > b$. When $\alpha \in [b, \beta]$, Algorithm 2 begins by setting $p_i = \frac{b}{\alpha + L}$ with a running length of $L + \alpha$ and then continues with $p_i = \frac{b}{U}$ for the second round with a running length of $L + \beta - L - \alpha$; when $\alpha \in [\beta, \delta]$, Algorithm 2 uses $p_i = \frac{b}{\alpha + L}$ with a running length of $L + \beta$ and stops; otherwise, the algorithm sets $p_i = \frac{b}{U}$ all the time. Therefore, the expected budget is

$$
\begin{aligned}
\mathbb{E}[\text{Budget}] &= \int_b^\beta \left[ \frac{b}{L+\alpha}(L+\alpha) + \frac{b}{U}(L+\beta-L-\alpha) \right] \frac{1}{\alpha} d\alpha + \int_\beta^\delta \frac{b}{L+\alpha}(L+\beta)\frac{1}{\alpha} d\alpha \\
&\quad + \int_\delta^{be} \frac{b}{U}(L+\beta)\frac{1}{\alpha} d\alpha \\
&= b\ln\frac{\beta}{b} + \frac{b\beta}{U}\ln\frac{\beta}{b} - \frac{b}{U}(\beta-b) + \frac{b(L+\beta)}{L}\left(\ln\frac{\delta}{\beta} - \ln\frac{L+\delta}{L+\beta}\right) + \frac{b(L+\beta)}{U}\ln\frac{be}{\delta} \\
&\leq b\ln\frac{U-L}{b} + \frac{b(U-L)}{U}\ln\frac{U-L}{b} - \frac{b}{U}(U-L-b) + \frac{bU}{U}\ln\frac{be}{\delta} \quad \text{increasing with } \beta \ (\beta = U-L) \\
&\leq b\ln(e-1) + \frac{b(b(e-1))}{L+b(e-1)}\ln(e-1) - \frac{b^2}{L+b(e-1)}(e-2) + b\ln\frac{be}{b(e-1)} \\
&\quad \text{increasing with } U \ (U = L + b(e-1)) \\
&\leq b\ln(e-1) + b\frac{e-1}{e}\ln(e-1) - \frac{b}{e}(e-2)) + b\ln\frac{e}{e-1} \quad \text{decreasing with } L \ (L = b) \\
&\leq b + b\left(\frac{e-1}{e}\ln(e-1) - \frac{e-2}{e}\right) \\
&\approx b
\end{aligned}
$$

### Subroutine 5

Similarly, we assume $\tau^* = L + \beta e^{j^*}$. Under the condition that $\delta \leq b(e+1)$, we have $j^* = 1$ or $j^* = 0$. As before, we only consider the worst case $j^* = 1$.

Let $\kappa = \frac{\delta}{e}$. Note that, when $\alpha \in [b, \kappa)$, Algorithm 2 first sets $p_i = \frac{b}{L+\alpha e}$ with a running length of $L + \alpha$, then transitions to $p_{k,i} = \frac{b}{U}$ with a running length of $L + \beta e - L - \alpha$. However, when $\alpha \in [\kappa, be]$, Algorithm 2 keeps setting $p_i = \frac{b}{U}$ with a

running length of $L + \beta e$. Therefore, the expected budget is

$$\mathbb{E}[\text{Budget}] = \int_b^\kappa \left[ \frac{b}{L+\alpha e}(L+\alpha) + \frac{b}{U}(L+\beta e - L - \alpha) \right] \frac{1}{\alpha} d\alpha + \int_\kappa^{be} \frac{b}{U}(L+\beta e) \frac{1}{\alpha} d\alpha$$

$$= b \left( \ln \frac{\kappa}{b} + (\frac{1}{e} - 1) \ln \frac{L+\kappa e}{L+be} \right) + \frac{b}{U} \beta e \ln \frac{\kappa}{b} - \frac{b}{U}(\kappa - b) + \frac{b(\beta e + L)}{U} \ln \frac{be}{\kappa}$$

$$\le b \left( \ln \frac{\kappa}{b} + (\frac{1}{e} - 1) \ln \frac{L+\kappa e}{L+be} \right) + \frac{b}{U}(U-L) \ln \frac{\kappa}{b} - \frac{b}{U}(\kappa - b) + b \ln \frac{be}{\kappa}$$

increasing with $\beta$ $(\beta = (U-L)/e)$

$$\le b + b(\frac{1}{e} - 1) \ln \frac{L+b(e+1)}{L+be} + \frac{b^2}{L+b(e+1)}(e+1) \ln \frac{e+1}{e} - \frac{b^2}{L+b(e+1)}(\frac{e+1}{e} - 1)$$

increasing with $U$ $(U = L + b(e+1))$

$$\le b + b(\frac{1-e}{e} \ln \frac{e+2}{e+1} + \frac{e+1}{e+2} \ln \frac{e+1}{e} - \frac{1}{e(e+2)}) \quad \text{decreasing with } L \ (L = b)$$

$$\approx b$$

## Subroutine 6

Under the assumption of $\tau^* = L + \beta e^{j^*}$, and given the condition $U > be^2$, it is possible for $j^*$ to be 0 or to extend to infinity. We only focus on the worst-case scenario, i.e., $j^* \ge 1$.

When $\alpha$ falls within $[b, \beta]$, Algorithm 2 stops with $p_i = b\left(1 - \frac{L+\alpha-b}{L+\alpha(e-1)}\right) \frac{(1-1/e)^{j^*}}{\alpha e^{j^*+2}}$ with a running length of $L + \beta e^{j^*} - L - \alpha e^{j^*}$. Otherwise, Algorithm 2 stops with $p_i = b\left(1 - \frac{L+\alpha-b}{L+\alpha(e-1)}\right) \frac{(1-1/e)^{j^*-1}}{\alpha e^{j^*+1}}$ with a running length of $L + \beta e^{j^*} - L - \alpha e^{j^*-1}$). Therefore, the expected budget is

$$\mathbb{E}[\text{Budget}] = \int_\beta^{be} \left[ \frac{b}{L+\alpha(e-1)}(L+\alpha) + b\left(1 - \frac{L+\alpha-b}{L+\alpha(e-1)}\right) \sum_{j=2}^{j^*} \frac{(1-1/e)^{j-2}}{\alpha e^j}(\alpha e^{j-1} - \alpha e^{j-2})\right.$$

$$+ b\left(1 - \frac{L+\alpha-b}{L+\alpha(e-1)}\right) \frac{(1-1/e)^{j^*-1}}{\alpha e^{j^*+1}}(L + \beta e^{j^*} - L - \alpha e^{j^*-1}) \Bigg] \frac{1}{\alpha}d\alpha$$

$$+ \int_b^\beta \left[ \frac{b}{L+\alpha(e-1)}(L+\alpha) + b\left(1 - \frac{L+\alpha-b}{L+\alpha(e-1)}\right) \sum_{j=2}^{j^*+1} \frac{(1-1/e)^{j-2}}{\alpha e^j}(\alpha e^{j-1} - \alpha e^{j-2})\right.$$

$$+ b\left(1 - \frac{L+\alpha-b}{L+\alpha(e-1)}\right) \frac{(1-1/e)^{j^*}}{\alpha e^{j^*+2}}(L + \beta e^{j^*} - L - \alpha e^{j^*}) \Bigg] \frac{1}{\alpha}d\alpha$$

$$\leq b\left(\ln\frac{be}{\beta} + (\frac{1}{e-1} - 1)\ln\frac{L+be(e-1)}{L+\beta(e-1)}\right) + b(1 - \frac{1}{e} - (1-\frac{1}{e})^{j^*})\frac{e-2}{e-1}\ln\frac{L+be(e-1)}{L+\beta(e-1)}$$

$$+ b(1 - \frac{1}{e} - (1-\frac{1}{e})^{j^*})\frac{b}{L}\left(\ln\frac{be}{\beta} - \frac{L+be(e-1)}{L+\beta(e-1)}\right)$$

$$+ b\beta e^{j^*}\frac{(e-1)^{j^*}}{e^{2j^*}}\frac{1}{L}\left(\ln\frac{be}{\beta} - \ln\frac{L+be(e-1)}{L+\beta(e-1)}\right) - b\frac{(e-1)^{j^*-1}}{e^{j^*+1}}\ln\frac{L+be(e-1)}{L+\beta(e-1)}$$

$$+ b\left(\ln\frac{\beta}{b} + (\frac{1}{e-1} - 1)\ln\frac{L+\beta(e-1)}{L+b(e-1)}\right) + b(1 - \frac{1}{e} - (1-\frac{1}{e})^{j^*+1})\frac{e-2}{e-1}\ln\frac{L+\beta(e-1)}{L+b(e-1)}$$

$$+ b(1 - \frac{1}{e} - (1-\frac{1}{e})^{j^*})\frac{b}{L}\left(\ln\frac{\beta}{b} - \frac{L+\beta(e-1)}{L+b(e-1)}\right)$$

$$+ b\beta e^{j^*}\frac{(e-1)^{j^*+1}}{e^{2j^*+2}}\frac{1}{L}\left(\ln\frac{\beta}{b} - \ln\frac{L+\beta(e-1)}{L+b(e-1)}\right) - b\frac{(e-1)^{j^*}}{e^{j^*+2}}\ln\frac{L+\beta(e-1)}{L+b(e-1)}$$

$$\leq b\left(1 + (\frac{1}{e-1} - 1)\ln\frac{L+be(e-1)}{L+b(e-1)}\right) + b(1 - \frac{1}{e} - (1-\frac{1}{e})^{j^*+1})\frac{e-2}{e-1}\ln\frac{L+be(e-1)}{L+b(e-1)}$$

$$+ b(1 - \frac{1}{e} - (1-\frac{1}{e})^{j^*})\frac{b}{L}\left(1 - \frac{L+be(e-1)}{L+b(e-1)}\right)$$

$$+ b^2\frac{(e-1)^{j^*+1}}{e^{j^*+1}}\frac{1}{L}\left(1 - \ln\frac{L+be(e-1)}{L+b(e-1)}\right) - b\frac{(e-1)^{j^*}}{e^{j^*+2}}\ln\frac{L+be(e-1)}{L+b(e-1)}$$

increasing with $\beta$ ($\beta = be$)

$$\leq b\left(1 + (\frac{1}{e-1} - 1)\ln\frac{L+be(e-1)}{L+b(e-1)}\right) + b\frac{e-2}{e}\ln\frac{L+be(e-1)}{L+b(e-1)}$$

$$+ b(1 - \frac{1}{e})\frac{b}{L}\left(1 - \ln\frac{L+be(e-1)}{L+b(e-1)}\right)$$

$$\approx b$$

Combining the above results with the proof of Subroutine 2, presented in Appendix C.1, establishes Lemma 4.1. □

**D.2. Proof of Theorem 4.2: Consistency and Robustness**

*Proof.* We begin with the proof of consistency and then proceed to the analysis of robustness.

**Consistency Analysis** It is straightforward to show that our algorithm is 1- consistent. When the width of the predictive interval is zero, meaning that $L = U = \tau^*$, we have

$$\mathbb{E}[\text{SOL}] = \frac{b}{U}\tau^* = b.$$

**Robustness Analysis** Below we show the robustness of our algorithm under each subroutine.

**Subroutine 4**

For cases where $\delta = U - L \leq b$, the algorithm proceeds with $p_{k,i} = \frac{b}{U}$. Hence, we have

$$\mathbb{E}[\text{SOL}] = \frac{b}{U}\beta \geq \frac{b}{L+b}L \geq \frac{b}{2}.$$

Next, we consider the case where $b(e-1) \geq \delta > b$, further divided into $\tau^* < L + b$ and $\tau^* \geq L + b$.

Suppose $\tau^* < L + b$. When $\alpha$ falls within $[b, \delta]$, Algorithm 2 assigns $p_i = \frac{b}{L+\alpha}$ with a running length of $\tau^*$. Otherwise, Algorithm 2 sets $p_i = \frac{b}{U}$ with a running length of $\tau^*$. Therefore, the expected solution is

$$\mathbb{E}[\text{SOL}] = \int_b^\delta \left[\frac{b}{L+\alpha}\tau^*\right]\frac{1}{\alpha}d\alpha + \int_\delta^{be}\frac{b}{U}\tau^*\frac{1}{\alpha}d\alpha$$

$$= \frac{b\tau^*}{L}\left(\ln\frac{\delta}{b} - \ln\frac{L+\delta}{L+b}\right) + \frac{b\tau^*}{U}\ln\frac{be}{\delta}$$

$$\geq b\left(\ln\frac{\delta}{b} - \ln\frac{L+\delta}{L+b}\right) + \frac{bL}{U}\ln\frac{be}{\delta} \quad \text{increasing with } \tau^* \ (\tau^* = L)$$

$$\geq b\left(\ln(e-1) - \ln\frac{L+b(e-1)}{L+b}\right) + \frac{bL}{L+b(e-1)}\ln\frac{e}{e-1}$$

$$\text{decreasing with } U \ (U = L + b(e-1))$$

$$\geq b\left(\ln(e-1) - \ln\frac{e}{2}\right) + b\frac{1}{e}\ln\frac{e}{e-1} \quad \text{increasing with } L \ (L = b)$$

$$= b\left(\ln\frac{2(e-1)}{e} + \frac{1}{e}\ln\frac{e}{e-1}\right)$$

For cases where $\tau^* \geq L + b$, let us suppose $\tau^* = L + \beta e^{j^*}$ where $\beta \in [b, be]$. Under the condition $U \leq be$ or $\delta \leq b(e-1)$, we have $j^* = 0$. Further, since $L + \beta \leq U$, we have $\beta \leq U - L$. When $\alpha$ falls within $[b, \beta]$, Algorithm 2 starts with $p_i = \frac{b}{L+\alpha}$ with a running length of $L + \alpha$, then transitions to $p_i = \frac{b}{U}$ with a running length of $L + \beta - L - \alpha$; when $\alpha$ falls within $[\beta, \delta]$, Algorithm 2 assigns $p_i = \frac{b}{L+\alpha}$ with a running length of $L + \beta$; otherwise, Algorithm 2 assigns $p_i = \frac{b}{U}$ with a running length of $L + \beta$. It follows that

$$\mathbb{E}[\text{SOL}] = \int_b^\beta \left[\frac{b}{L+\alpha}(L+\alpha) + \frac{b}{U}(L+\beta-L-\alpha) - \sigma\ln\frac{U}{L+\alpha}\right]\frac{1}{\alpha}d\alpha$$

$$+ \int_\beta^\delta \frac{b}{L+\alpha}(L+\beta)\frac{1}{\alpha}d\alpha + \int_\delta^{be}\frac{b}{U}(L+\beta)\frac{1}{\alpha}d\alpha$$

$$= b\ln\frac{\beta}{b} + \frac{b\beta}{U}\ln\frac{\beta}{b} - \frac{b}{U}(\beta-b) - \sigma\ln\frac{e}{2}\ln\frac{\beta}{b} + \frac{b(L+\beta)}{L}\left(\ln\frac{\delta}{\beta} - \ln\frac{L+\delta}{L+\beta}\right) + \frac{b(L+\beta)}{U}\ln\frac{be}{\delta}$$

$$\geq \frac{b(L+b)}{L}\left(\ln\frac{U-L}{b} - \ln\frac{U}{L+b}\right) + \frac{b(L+b)}{U}\ln\frac{be}{U-L} \quad \text{increasing with } \beta \ (\beta = b)$$

$$\geq \frac{b(L+b)}{L}\left(\ln(e-1) - \ln\frac{L+b(e-1)}{L+b}\right) + \frac{b(L+b)}{L+b(e-1)}\ln\frac{e}{e-1}$$

$$\text{decreasing with } U \ (U = L + b(e-1))$$

$$\geq 2b\ln\frac{2(e-1)}{e} + b\frac{2}{e}\ln\frac{e}{e-1} \quad \text{increasing with } L \ (L = b)$$

$$= b(2\ln\frac{2(e-1)}{e} + \frac{2}{e}\ln\frac{e}{e-1})$$

**Subroutine 5**

For the situation where $\delta = U - L \leq be$, the algorithm's procedure involves setting $p_i = \frac{b}{U}$ for all time points. Hence, we

have

$$\mathbb{E}[\text{SOL}] = \frac{b}{U}\tau^* \geq \frac{b}{L + be}L \geq \frac{b}{e + 1}.$$

Next, we consider the case where $b(e + 1) \geq \delta > be$, further divided into $\tau^* < L + b$ and $\tau^* \geq L + b$.

First, suppose $\tau^* < L + b$. When $\alpha$ falls in the range $[b, \kappa]$, Algorithm 2 assigns $p_i = \frac{b}{L + \alpha e}$ with a running length $\tau^*$. Otherwise, Algorithm 2 assigns $p_i = \frac{b}{U}$ with a running length $\tau^*$. Therefore, the expected solution is

$$
\begin{aligned}
\mathbb{E}[\text{SOL}] &= \int_b^\kappa \left[ \frac{b}{L + \alpha e}\tau^* \right] \frac{1}{\alpha}d\alpha + \int_\kappa^{be} \frac{b}{U}\tau^* \frac{1}{\alpha}d\alpha \\
&= \frac{b\tau^*}{L}\left( \ln\frac{\kappa}{b} - \ln\frac{L + \kappa e}{L + be} \right) + \frac{b\tau^*}{U}\ln\frac{be}{\kappa} \\
&\geq b\left( \ln\frac{\kappa}{b} - \ln\frac{L + \kappa e}{L + be} \right) + \frac{bL}{U}\ln\frac{be}{\kappa} \quad \text{increasing with } \tau^* \ (\tau^* = L) \\
&\geq b\left( \ln\frac{e + 1}{e} - \ln\frac{L + b(e + 1)}{L + be} \right) + \frac{bL}{L + b(e + 1)}\ln\frac{e^2}{e + 1} \\
&\quad \text{decreasing with } U \ (U = L + b(e + 1)) \\
&\geq b\left( \ln\frac{e + 1}{e} - \ln\frac{e^2}{e^2 - 1} \right) + b\frac{e^2 - e - 1}{e^2}\ln\frac{e^2}{e + 1} \\
&\quad \text{increasing with } L \ (L = b(e^2 - e - 1)) \\
&= b\left( \ln\frac{e + 1}{e} - \ln\frac{e^2}{e^2 - 1} + \frac{e^2 - e - 1}{e^2}\ln\frac{e^2}{e + 1} \right)
\end{aligned}
$$

For situations where $\tau^* \geq L + b$, suppose $\tau^* = L + \beta e^{j^*}$ where $\beta \in [b, be]$. Below we separately consider two cases: 1) $j^* \geq 1$ or $\beta \geq \kappa$, and 2) $j^* = 0$ and $\beta < \kappa$.

Suppose case 1) where $j^* \geq 1$ or $\beta \geq \kappa$. When $\alpha$ falls within $[b, \kappa]$, Algorithm 2 first assigns $p_i = \frac{b}{L + \alpha e}$ for a time length of $L + \alpha$, and then proceeds with $p_{k,i} = \frac{b}{U}$ with a running length $L + \beta e^{j^*} - L - \alpha$. Otherwise, Algorithm 2 assigns

$p_i = \frac{b}{U}$ with a running length of $L + \beta e^{j^*}$. Therefore, the expected solution is

$$
\mathbb{E}[\text{SOL}] = \int_b^\kappa \left[ \frac{b}{L + \alpha e}(L + \alpha) + \frac{b}{U}(\beta e^{j^*} + L - L - \alpha) - \sigma \ln \frac{U}{L + \alpha e} \right] \frac{1}{\alpha} d\alpha
$$

$$
+ \int_\kappa^{be} \frac{b}{U}(\beta e^{j^*} + L) \frac{1}{\alpha} d\alpha
$$

$$
\geq b \left( \ln \frac{\kappa}{b} + (\frac{1}{e} - 1) \ln \frac{L + \kappa e}{L + be} \right) + \frac{b}{U}\beta e^{j^*} \ln \frac{\kappa}{b} - \frac{b}{U}(\kappa - b) - \sigma \ln \frac{e+2}{e+1} \ln \frac{\kappa}{b}
$$

$$
+ \frac{b(\beta e^{j^*} + L)}{U} \ln \frac{be}{\kappa}
$$

$$
\geq b \left( \ln \frac{\kappa}{b} + (\frac{1}{e} - 1) \ln \frac{L + \kappa e}{L + be} \right) + \frac{b}{U}\kappa \ln \frac{\kappa}{b} - \frac{b}{U}(\kappa - b) - \sigma \ln \frac{e+2}{e+1} \ln \frac{\kappa}{b}
$$

$$
+ \frac{b(\kappa + L)}{U} \ln \frac{be}{\kappa} \quad \text{increasing with } \beta, j^* \ (\beta = \kappa, j^* = 0)
$$

$$
\geq b \left( \ln \frac{e+1}{e} - \frac{e-1}{e} \ln \frac{L + b(e+1)}{L + be} \right) + \frac{b^2(e+1)}{Le + be(e+1)} \ln \frac{e+1}{e} - \frac{b}{L + b(e+1)}(\frac{b(e+1)}{e} - b)
$$

$$
- \sigma \ln \frac{e+2}{e+1} \ln \frac{e+1}{e} + \frac{b(L + \frac{b(e+1)}{e})}{L + b(e+1)} \ln \frac{e^2}{e+1} \quad \text{decreasing with } U \ (U = L + b(e+1))
$$

$$
\geq b(\ln \frac{e+1}{e} - \frac{e-1}{e} \ln \frac{e^2}{e^2 - 1}) + \frac{b(e+1)}{e^3} \ln \frac{e+1}{e} - b\frac{1}{e^2}\frac{1}{e} - \sigma \ln \frac{e+2}{e+1} \ln \frac{e+1}{e}
$$

$$
+ b\frac{e^2 - e - 1 + \frac{(e+1)}{e}}{e^2} \ln \frac{e^2}{e+1} \quad \text{increasing with } L \ (L = b(e^2 - e - 1))
$$

$$
= b \left( \ln \frac{e+1}{e} - \frac{1}{e^3} - \frac{e-1}{e} \ln \frac{e^2}{e^2 - 1} + \frac{e^2 - e - 1}{e^2} \ln \frac{e^2}{e+1} \right) + b\frac{1 + \frac{1}{e}}{e^2}
$$

$$
- \sigma \ln \frac{e+2}{e+1} \ln \frac{e+1}{e}
$$

$$
= b \left( (1 + \frac{1}{e^2}) \ln(e+1) - 1 - \frac{1}{e^2} + \frac{e-1}{e} \ln(e-1) \right) - \sigma \ln \frac{e+2}{e+1} \ln \frac{e+1}{e}
$$

Next, consider case 2) where $j^* = 0$ and $\beta < \kappa$. When $\alpha$ falls within $[b, \beta]$, Algorithm 2 starts with $p_i = \frac{b}{L + \alpha e}$ with a running length of $L + \alpha$, then transits to $p_i = \frac{b}{U}$ with a running length of $L + \beta - L - \alpha$; when $\alpha \in [\beta, \kappa]$, Algorithm 2 sets $p_i = \frac{b}{L + \alpha e}$ with a running length of $L + \beta$; otherwise, Algorithm 2 sets $p_i = \frac{b}{U}$ with a running length of $L + \beta$. Therefore,

we have

$$\mathbb{E}[\text{SOL}] = \int_b^\beta \left[ \frac{b}{L + \alpha e}(L + \alpha) + \frac{b}{U}(\beta + L - L - \alpha) - \sigma \ln \frac{U}{L + \alpha e} \right] \frac{1}{\alpha} d\alpha$$

$$+ \int_\beta^\kappa \frac{b}{L + \alpha e}(L + \beta)\frac{1}{\alpha}d\alpha + \int_\kappa^{be} \frac{b}{U}(L + \beta)\frac{1}{\alpha}d\alpha$$

$$= b \left( \ln \frac{\beta}{b} + (\frac{1}{e} - 1) \ln \frac{L + \beta e}{L + be} \right) + \frac{b}{U}\beta \ln \frac{\beta}{b} - \frac{b}{U}(\beta - b) - \sigma \ln \frac{e^2}{e^2 - 1} \ln \frac{\beta}{b}$$

$$+ \frac{b(L + \beta)}{L} \left( \ln \frac{\kappa}{\beta} - \ln \frac{L + \kappa e}{L + \beta e} \right) + \frac{b(L + \beta)}{U} \ln \frac{be}{\kappa}$$

$$\geq \frac{b(L + \beta)}{L} \left( \ln \frac{\kappa}{\beta} - \ln \frac{L + \kappa e}{L + \beta e} \right) + \frac{b(L + \beta)}{U} \ln \frac{be}{\kappa} \quad \text{increasing with } \beta \ (\beta = b)$$

$$\geq \frac{b(L + b)}{L} \left( \ln \frac{b(e + 1)}{be} - \ln \frac{L + b(e + 1)}{L + be} \right) + \frac{b(L + b)}{L + b(e + 1)} \ln \frac{e^2}{e + 1}$$

$$\text{decreasing with } U \ (U = L + b(e + 1))$$

$$\geq b\frac{e^2 - e}{e^2 - e - 1} \left( \ln \frac{e + 1}{e} - \ln \frac{e^2}{e^2 - 1} \right) + \frac{e^2 - e}{e^2} \ln \frac{e^2}{e + 1}$$

$$\text{increasing with } L \ (L = b(e^2 - e - 1))$$

$$\geq b \left( \frac{e^2 - e}{e^2 - e - 1} \ln \frac{(e + 1)^2(e - 1)}{e^3} + \frac{e - 1}{e} \ln \frac{e^2}{e + 1} \right).$$

## Subroutine 6

In this scenario, our algorithm initiates with $p_{k,i} = \frac{b}{L + \alpha(e-1)}$, subsequently updating $\tilde{\tau}$ and $b$ after each iteration. We analyze two cases: one where $\tau^* < L + b$, and the other where $\tau^* \geq L + b$.

For the first situation where $\tau^* < L + b$, Algorithm 2 consistently sets $p_i = \frac{b}{L + \alpha(e-1)}$. Therefore, we have

$$\mathbb{E}[\text{SOL}] = \int_b^{be} \frac{b}{L + \alpha(e-1)}\tau^*\frac{1}{\alpha}d\alpha$$

$$= \frac{b\tau^*}{L} \left( \ln \frac{be}{b} - \ln \frac{L + be(e-1)}{L + b(e-1)} \right)$$

$$\geq \tau^* \left( 1 - \ln \frac{e^2 - e + 1}{e} \right) \quad \text{increasing with } L \ (L = b)$$

$$\geq b \left( 2 - \ln(e^2 - e + 1) \right)$$

Next, we consider the case where $\tau^* \geq L + b$. Suppose that $\tau^* = L + \beta e^{j^*}$ where $\beta \in [b, be]$.

When $j^* \geq 1$, the objective function when $\alpha \geq \beta$ is

$$\text{SOL}_1 = \frac{b}{L + \alpha(e-1)}(L + \alpha) + b \left( 1 - \frac{L + \alpha - b}{L + \alpha(e-1)} \right) \sum_{j=2}^{j^*} \frac{(1 - 1/e)^{j-2}}{\alpha e^j}(\alpha e^{j-1} - \alpha e^{j-2})$$

$$+ b \left( 1 - \frac{L + \alpha - b}{L + \alpha(e-1)} \right) \frac{(1 - 1/e)^{j^*-1}}{\alpha e^{j^*}}(L + \beta e^{j^*} - L - \alpha e^{j^*-1})$$

$$- \sigma \ln \frac{\alpha e^{2j^*+1}}{(\alpha(e-2) + b)(e-1)^{j^*-1}}$$

The objective function when $\alpha < \beta$ is

$$
\begin{aligned}
\text{SOL}_2 &= \frac{b}{L + \alpha(e-1)}(L + \alpha) + b\left(1 - \frac{L + \alpha - b}{L + \alpha(e-1)}\right) \sum_{j=2}^{j^*+1} \frac{(1 - 1/e)^{j-2}}{\alpha e^j}(\alpha e^{j-1} - \alpha e^{j-2}) \\
&\quad + b\left(1 - \frac{L + \alpha - b}{L + \alpha(e-1)}\right) \frac{(1 - 1/e)^{j^*}}{\alpha e^{j^*+2}}(L + \beta e^{j^*} - L - \alpha e^{j^*}) \\
&\quad - \sigma \ln \frac{\alpha e^{2j^*+2}}{(\alpha(e-2) + b)(e-1)^{j^*}}.
\end{aligned}
$$

The expected solution is

$$\mathbb{E}[\text{SOL}] = \int_{\beta}^{be} \text{SOL}_1 \frac{1}{\alpha} d\alpha + \int_{b}^{\beta} \text{SOL}_2 \frac{1}{\alpha} d\alpha$$

$$\geq b \left( \ln \frac{be}{\beta} + (\frac{1}{e-1} - 1) \ln \frac{L + be(e-1)}{L + \beta(e-1)} \right) + b(1 - \frac{1}{e} - (1 - \frac{1}{e})^{j^*}) \frac{e-2}{e-1} \ln \frac{L + be(e-1)}{L + \beta(e-1)}$$

$$+ b^2(1 - \frac{1}{e} - (1 - \frac{1}{e})^{j^*}) \frac{1}{L} \left( \ln \frac{be}{\beta} - \ln \frac{L + be(e-1)}{L + \beta(e-1)} \right)$$

$$+ b\beta e^{j^*} \frac{(e-1)^{j^*-1}}{e^{2j^*}} \frac{e-2}{L} \left( \ln \frac{be}{\beta} - \ln \frac{L + be(e-1)}{L + \beta(e-1)} \right) - b \frac{(e-1)^{j^*-1}}{e^{j^*+1}} \frac{e-2}{e-1} \ln \frac{L + be(e-1)}{L + \beta(e-1)}$$

$$- \sigma \ln \frac{e^{2j^*+1}}{(e-1)^{j^*+1}} \ln \frac{be}{\beta}$$

$$+ b \left( \ln \frac{\beta}{b} + (\frac{1}{e-1} - 1) \ln \frac{L + \beta(e-1)}{L + b(e-1)} \right) + b(1 - \frac{1}{e} - (1 - \frac{1}{e})^{j^*+1}) \frac{e-2}{e-1} \ln \frac{L + \beta(e-1)}{L + b(e-1)}$$

$$+ b^2(1 - \frac{1}{e} - (1 - \frac{1}{e})^{j^*+1}) \frac{1}{L} \left( \ln \frac{\beta}{b} - \ln \frac{L + \beta(e-1)}{L + b(e-1)} \right)$$

$$+ b\beta e^{j^*} \frac{(e-1)^{j^*}}{e^{2j^*+2}} \frac{e-2}{L} \left( \ln \frac{\beta}{b} - \ln \frac{L + \beta(e-1)}{L + b(e-1)} \right) - b \frac{(e-1)^{j^*}}{e^{j^*+2}} \frac{e-2}{e-1} \ln \frac{L + \beta(e-1)}{L + b(e-1)}$$

$$- \sigma \ln \frac{e^{2j^*+3}}{(e-1)^{j^*+2}} \ln \frac{\beta}{b}$$

$$\geq b \left( 1 + (\frac{1}{e-1} - 1) \ln \frac{L + be(e-1)}{L + b(e-1)} \right) + b(1 - \frac{1}{e} - (1 - \frac{1}{e})^{j^*}) \frac{e-2}{e-1} \ln \frac{L + be(e-1)}{L + b(e-1)}$$

$$+ b^2(1 - \frac{1}{e} - (1 - \frac{1}{e})^{j^*}) \frac{1}{L} \left( 1 - \ln \frac{L + be(e-1)}{L + b(e-1)} \right)$$

$$+ b^2 e^{j^*} \frac{(e-1)^{j^*-1}}{e^{2j^*}} \frac{e-2}{L} \left( 1 - \ln \frac{L + be(e-1)}{L + b(e-1)} \right) - b \frac{(e-1)^{j^*-1}}{e^{j^*+1}} \frac{e-2}{e-1} \ln \frac{L + be(e-1)}{L + b(e-1)}$$

$$- \sigma \ln \frac{e^{2j^*+1}}{(e-1)^{j^*+1}} \quad \text{increasing with } \beta \ (\beta = b)$$

$$\geq b \left( 1 + (\frac{1}{e-1} - 1) \ln \frac{b(e^2 - e + 1)}{b + b(e-1)} \right) + b(1 - \frac{1}{e} - (1 - \frac{1}{e})^{j^*}) \frac{e-2}{e-1} \ln \frac{b(e^2 - e + 1)}{b + b(e-1)}$$

$$+ b(1 - \frac{1}{e} - (1 - \frac{1}{e})^{j^*}) \left( 1 - \ln \frac{e^2 - e + 1}{e} \right)$$

$$+ b e^{j^*} \frac{(e-1)^{j^*-1}}{e^{2j^*}} (e-2) \left( 1 - \ln \frac{b(e^2 - e + 1)}{b + b(e-1)} \right) - b \frac{(e-1)^{j^*-1}}{e^{j^*+1}} \frac{e-2}{e-1} \ln \frac{e^2 - e + 1}{e}$$

$$- \sigma \ln \frac{e^{2j^*+1}}{(e-1)^{j^*+1}} \quad \text{increasing with } L \ (L = b)$$

$$= b \left( 1 + (\frac{1}{e-1} - 1) \ln \frac{e^2 - e + 1}{e} \right) + b(1 - \frac{1}{e} - (1 - \frac{1}{e})^{j^*}) \frac{e-2}{e-1} \ln \frac{e^2 - e + 1}{e}$$

$$+ b(1 - \frac{1}{e} - (1 - \frac{1}{e})^{j^*}) \left( 1 - \ln \frac{e^2 - e + 1}{e} \right)$$

$$+ b \frac{(e-1)^{j^*-1}}{e^{j^*}} (e-2) \left( 1 - \ln \frac{e^2 - e + 1}{e} \right) - b \frac{(e-1)^{j^*-1}}{e^{j^*+1}} \frac{e-2}{e-1} \ln \frac{e^2 - e + 1}{e}$$

$$- \sigma \ln \frac{e^{2j^*+1}}{(e-1)^{j^*+1}}.$$

Next, consider the case where $j^* = 0$. When $\alpha \in [b, \beta]$, Algorithm 2 starts with $p_i = \frac{b}{L + \alpha(e-1)}$ with a running length of

$L + \alpha$, then transitions to $p_i = b\left(1 - \frac{L+\alpha-b}{L+\alpha(e-1)}\right)\frac{1}{\alpha e^2}$ with a running length of $L + \beta - L - \alpha$. Otherwise, Algorithm 2 consistently sets $p_i = \frac{b}{L+\alpha(e-1)}$. Therefore, we have

$$
\begin{aligned}
\mathbb{E}[\text{SOL}] &= \int_b^\beta \left[ \frac{b}{L + \alpha(e-1)}(L+\alpha) + b\left(1 - \frac{L+\alpha-b}{L+\alpha(e-1)}\right)\frac{1}{\alpha e^2}(L+\beta-L-\alpha) \right. \\
&\quad \left. - \sigma \ln \frac{\alpha e^2}{\alpha(e-2)+b} \right]\frac{1}{\alpha}d\alpha + \int_\beta^{be} \frac{b}{L+\alpha(e-1)}(L+\beta)\frac{1}{\alpha}d\alpha \\
&\geq b\left(\ln\frac{\beta}{b} - \frac{e-2}{e-1}\ln\frac{L+\beta(e-1)}{L+b(e-1)}\right) - \frac{b\beta}{e^2}\frac{e-2}{L}\left(\ln\frac{\beta}{b} - \ln\frac{L+\beta(e-1)}{L+b(e-1)}\right) \\
&\quad - \frac{b}{e^2}\frac{e-2}{e-1}\ln\frac{L+\beta(e-1)}{L+b(e-1)} - \sigma\ln\frac{e^3}{(e-1)^2}\ln\frac{\beta}{b} + \frac{b(L+\beta)}{L}\left(\ln\frac{be}{\beta} - \ln\frac{L+be(e-1)}{L+\beta(e-1)}\right) \\
&\geq \frac{b(L+b)}{L}\left(1 - \ln\frac{L+be(e-1)}{L+b(e-1)}\right) \quad \text{increasing with } \beta \ (\beta = b) \\
&\geq 2b\left(1 - \ln\frac{b(e^2-e+1)}{b+b(e-1)}\right) \quad \text{increasing with } L \ (L = b) \\
&= 2b\left(2 - \ln(e^2-e+1)\right).
\end{aligned}
$$

**Tuning parameter selection**  For Scenario 1) where $U \leq be$, the competitive ratio is

$$
\ln\frac{2(e-1)}{e} + \frac{1}{e}\ln\frac{e}{e-1}.
$$

For Scenario 2) where $be < U \leq be^2$, the competitive ratio is

$$
\min\left(\frac{1}{e}, \frac{2}{e} - \frac{\sigma}{b}\right).
$$

For Scenario 3) where $U > be^2$, the competitive ratio is

$$
\begin{aligned}
\min\left(2 - \ln(e^2-e+1), 1 + \left(\frac{1}{e-1} - 1\right)\ln\frac{e^2-e+1}{e} + (1 - \frac{1}{e} - (1 - \frac{1}{e})^{j^*})\frac{e-2}{e-1}\ln\frac{e^2-e+1}{e}\right. \\
+ (1 - \frac{1}{e} - (1 - \frac{1}{e})^{j^*})\left(1 - \ln\frac{e^2-e+1}{e}\right) + \frac{(e-1)^{j^*-1}}{e^{j^*}}(e-2)\left(1 - \ln\frac{e^2-e+1}{e}\right) \\
\left. - \frac{(e-1)^{j^*-1}}{e^{j^*+1}}\frac{e-2}{e-1}\ln\frac{e^2-e+1}{e} - \frac{\sigma}{b}\ln\frac{e^{2j^*+1}}{(e-1)^{j^*+1}}\right).
\end{aligned}
$$

By restricting the value of $\sigma$ under each scenario and combining the above results, we establish Theorem 4.2. Specifically, when $\sigma = \frac{1}{\tau^*}$, it can be verified that Theorem 4.2 holds.  □

# E. Additional Synthetic Experiments

### E.1. Performance under Small $\tau^*$

In this section, we examine the performances of the algorithms under the learning-augmented setting where $\tau^*$ is small. Specifically, we set the number of risk occurrences $\tau^* = \text{Int}[0.2(T+b)]$ for scenarios with horizon lengths $T = 22$ and $T = 100$, and $\tau^* = \text{Int}[0.1(T+b)]$ for scenario $T = 100$. Figure 4 presents the average competitive ratio against a range of prediction interval widths.

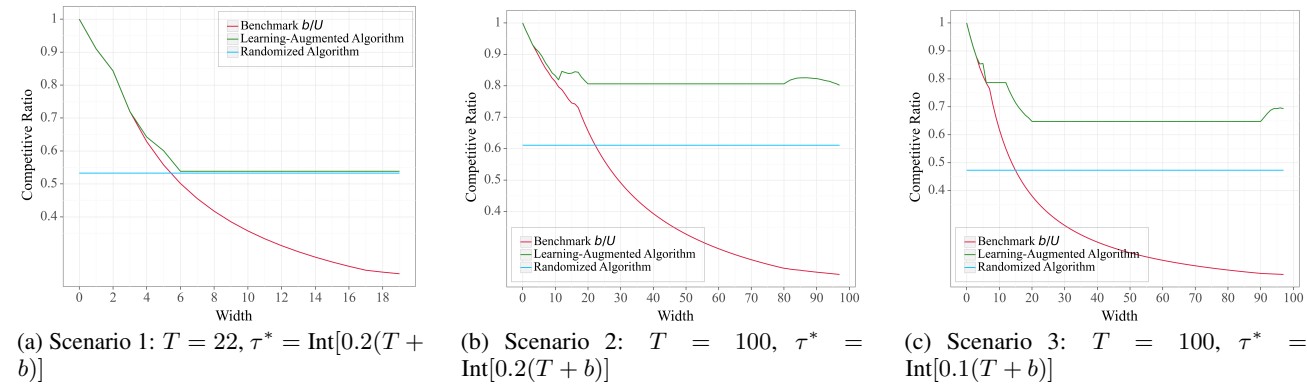

(a) Scenario 1: $T = 22$, $\tau^* = \text{Int}[0.2(T + b)]$

(b) Scenario 2: $T = 100$, $\tau^* = \text{Int}[0.2(T + b)]$

(c) Scenario 3: $T = 100$, $\tau^* = \text{Int}[0.1(T + b)]$

*Figure 4.* Average competitive ratio under learning-augmented setting with $b = 3$.

## E.2. Budget Utilization by Each Algorithm

To assess the budget utilization by each algorithm, we eliminate the penalty term from the objective in Problem 1. Figures 5 and 6 display the average competitive ratios in scenarios without and with learning augmentation, respectively. We note that in Figure 5 (middle), when $\tau^* = 22$, the competitive ratio slightly exceeds 1. This is attributed to our algorithm utilizing a slightly higher budget in expectation. We provide detailed insights into this observation in Section 1 of the Supplementary Material, where we demonstrate that the worst-case budget spent is about $1.047b_k$, slightly surpassing the allocated budget.

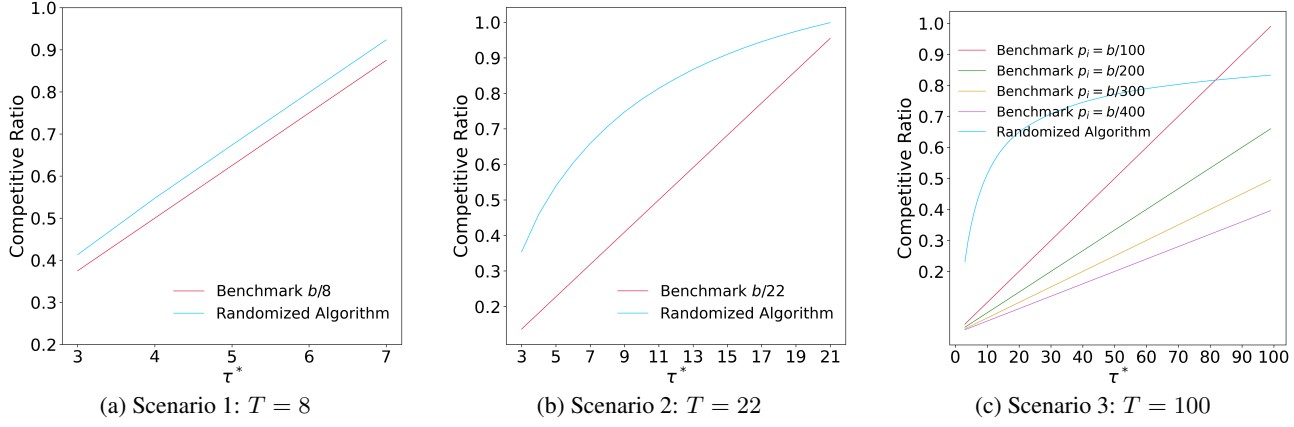

(a) Scenario 1: $T = 8$

(b) Scenario 2: $T = 22$

(c) Scenario 3: $T = 100$

*Figure 5.* Average competitive ratio under non-learning augmented setting with $b = 3$.

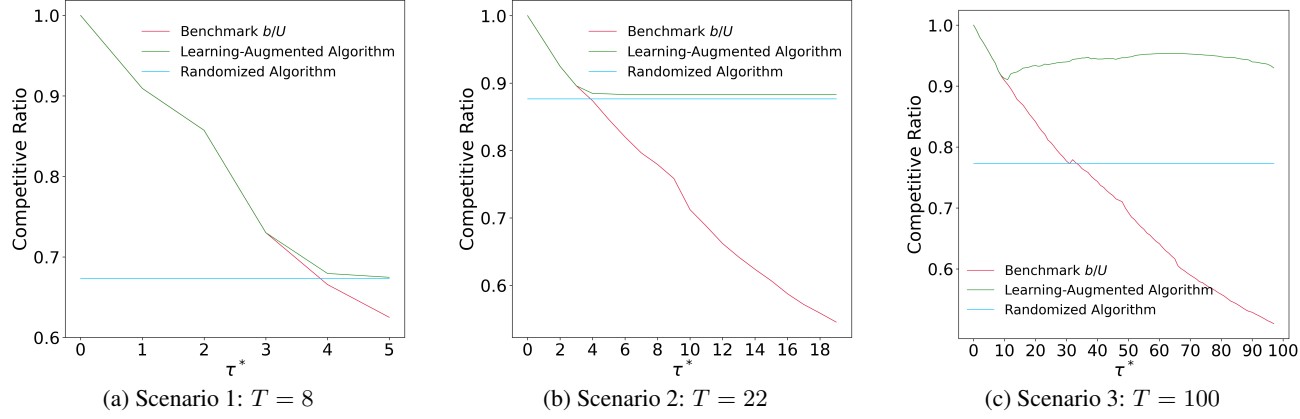

(a) Scenario 1: $T = 8$

(b) Scenario 2: $T = 22$

(c) Scenario 3: $T = 100$

*Figure 6.* Average competitive ratio under learning-augmented setting with $b = 3$.

## F. Additional Results on HeartSteps V1 Study

Our research is inspired by the Heartsteps V1 mobile health study, which aims to enhance physical activity among sedentary individuals (Klasnja et al., 2019). The study involved 37 participants over a follow-up period of six weeks, gathering detailed data on step counts on a minute-by-minute basis. To ensure the reliability of the step count data, our analysis was restricted to the hours from 9 am to 9 pm, with a decision time frequency set at five-minute intervals (Liao et al., 2018). This led to the accumulation of 1585 instances of 12-hour user-days, with $T = 144$ decision times per day.

At each decision time $t$, we define the risk variable $R_t$ with a binary classification: $R_t = 1$ indicates a sedentary state, identified by recording fewer than 150 steps in the prior 40 minutes, and $R_t = 0$ signifies a non-sedentary state. Additionally, the availability for intervention, $I_t$, is contingent on recent messaging activity: if the user has received an anti-sedentary message within the preceding hour, $I_t$ is set to 0; otherwise, it is set to 1. We want to distribute $b = 1.5$ interventions over available sedentary times each day.

We implement four algorithms: our randomized and learning-augmented algorithms (Algorithms 1 and 2, respectively), the SeqRTS strategy proposed by Liao et al. (2018), and a benchmark method ($b/U$). Rather than devising a tailored prediction model, we generate prediction intervals by randomly selecting from a range of $[2, 144]$, which contains $\tau^*$, with intervals of varying widths. This approach allows us to assess the performance of different algorithms under varying qualities of forecast accuracy.

We adopt the SeqRTS method to include prediction intervals, ensuring a balanced comparison with our algorithms. At the start of each user day, a number is randomly selected from the interval $[L, U]$ to estimate the number of available risk times. Should the budget be exhausted before allocating for all available risk times, a minimum probability of $1 \times 10^{-6}$ is assigned to the remaining times. For additional information on the SeqRTS method, readers are referred to Liao et al. (2018).

Figure 7 illustrates the average entropy change across user days. It is evident that SeqRTS exhibits the highest entropy change, suggesting non-uniform distribution behavior. In contrast, our learning-augmented algorithm demonstrates superior uniformity, outperforming the randomized algorithm. The benchmark method records an entropy of zero, attributed to its conservative strategy of assigning a constant probability of $b/U$.

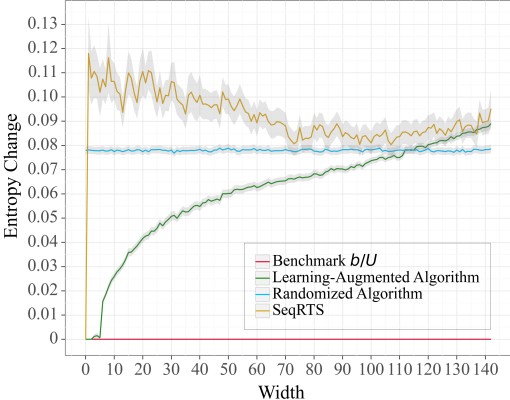

*Figure 7.* Average entropy change across user days under various prediction interval widths on HeartSteps V1 dataset. The shaded area indicates the $\pm 1.96$ standard error bounds across user days.

Figure 8 shows the average competitive ratio and entropy change across user days, considering the scenario where SeqRTS assigns a minimum probability of 0 to remaining risk times once the budget is depleted. Owing to the Penalization term 2, this results in the objective function being negative infinity and the entropy change reaching infinity.

## G. Derivation of Lower Bound

In this section, we derive a loose upper bound for any randomized algorithm for the OUS problem.

*Proof.* We utilize Yao's Lemma (Yao, 1977), which states that an upper bound can be established by constructing a distribution over problem instances where every deterministic algorithm performs poorly. We construct a randomized

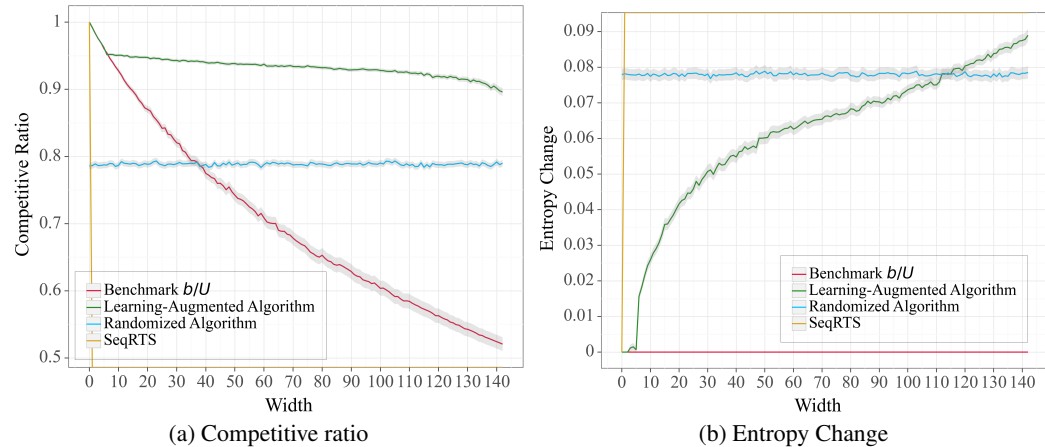

(a) Competitive ratio  (b) Entropy Change

*Figure 8.* Average competitive ratio and entropy change across user days under various prediction interval widths on the HeartSteps V1 dataset. The shaded area represents the $\pm 1.96$ standard error bounds across user days. *Note*: For SeqRTS, a minimum probability of 0 is assigned to the remaining times when the budget is exhausted.

instance $I$ with budget $b = 1$ and time horizon $T = 5$, where the true number of risk times $\tau^*$ takes values in $1, 2, 3, 4, 5$ with probabilities $\pi_1 = 0.6$, $\pi_2 = 0.15$, $\pi_3 = 0.1$, $\pi_4 = 0.1$, and $\pi_5 = 0.05$.

For this instance, the best deterministic algorithm with probabilities $(p_1, p_2, p_3, p_4, p_5)$ solves:

$$\arg \max_{p_1, p_2, p_3, p_4, p_5} \pi_1 p_1 + \pi_2(p_1 + p_2) + \pi_3(p_1 + p_2 + p_3) + \pi_4(p_1 + p_2 + p_3 + p_4) + \pi_5 b$$

$$- \pi_2 \frac{1}{2} \ln \frac{p_1}{p_2} - \pi_3 \frac{1}{3} \ln \frac{p_1}{p_3} - \pi_4 \frac{1}{4} \ln \frac{p_1}{p_4} - \pi_5 \frac{1}{5} \ln \frac{p_1}{p_5}$$

subject to $p_1 + p_2 + p_3 + p_4 + p_5 = b$.

The optimal solution yields probabilities $p_1 = 0.6970$, $p_2 = 0.1919$, $p_3 = 0.0606$, $p_4 = 0.0404$, and $p_5 = 0.0101$, achieving an expected competitive ratio of $0.504$. By Yao's Lemma, this implies no randomized algorithm can achieve a competitive ratio exceeding $0.504$. □

While this bound provides insight, deriving a tight upper bound remains an open challenge. The difficulty stems from several factors:

1. The analysis requires evaluating every possible deterministic algorithm's expected performance under the budget constraint.

2. Each deterministic algorithm is characterized by a sequence of randomization probabilities $(p_1, \ldots, p_T)$, making general analysis without a specific sequence structure intractable.

3. The objective function's non-smooth nature and the problem's distinct regimes (finite vs. infinite horizons) further complicate the analysis.

We leave the derivation of a tighter bound as future work, noting that our current bound suggests the potential existence of improved algorithms.

