# OpenReview forum: "Online Uniform Sampling: Randomized Learning-Augmented Approximation Algorithms with Application to Digital Health"
_ICML.cc/2025/Conference — Submitted to ICML 2025_

### Official Review · Reviewer_Nntf · 2025-02-24

**Overall Recommendation:** 4

**Summary:**

The authors propose an algorithm for online uniform sampling (OUS) to distribute a constrained sampling budget across unknown decision times as uniformly as possible over risk times. They consider cases of whether the number of risk times is both known and unknown, and present algorithms for both scenarios that are supported by both theoretical and empirical results.

**Claims And Evidence:**

To the best of my knowledge, I believe the claims made in the submission are supported by clear and convincing evidence. The authors cite relevant work, derive reasonable theoretical results, and present empirical findings that support the claims made in the paper. Regarding the claim in lines 208-211, left column, I am unable to validate this claim and defer to more knowledgeable reviewers.

**Essential References Not Discussed:**

The references included by the authors are extensive and seem reasonable to me. I do not currently do research in this space and am not well-versed enough with existing literature in this research area to comment on if any references are missing.

**Experimental Designs Or Analyses:**

I have checked for the soundness of the experimental design and analysis of both the synthetic and real-world experiments presented, and believe that they are sound to the best of my knowledge.

**Methods And Evaluation Criteria:**

Both the synthetic and mHealth datasets make sense for the problem at hand. Given the lack of baselines in the OUS research field as stated by the authors, the authors could potentially run their method on additional relevant dataset(s) and tasks (e.g., some options are listed [here](https://depressioncenter.org/research-services/mobile-technologies-core/mobile-health-datasets)) to better demonstrate the empirical efficacy of their method, although I would still recommend acceptance of this work as is without such additional experiments.

**Other Comments Or Suggestions:**

- [Line 332, right column] “inAlgorithm 2” is missing a space.
- Should the x-axis correspond to user-days instead of “Width” in Figure 3?

**Other Strengths And Weaknesses:**

## Strengths

2. In general, this is a well-written manuscript with notable clarity and presentation - I do not personally do research in this field, yet was able to follow the contributions and mathematical formulation of this paper quite easily even on the first pass.

## Weaknesses

In general, the "weaknesses" listed below are more so clarifying questions for myself.

3. Is the assumption for binary risk levels (line 85, right column) commonly used in the literature? How might the framework proposed in this paper extend to the continuous risk setting, which could offer a more descriptive picture of patient state? I appreciate the discussion of the extension to multiple risk levels by the authors in Appendix A, although would like to see empirical results to support the practical tractability of this extension to approximate a pseudo-continuous risk setting (e.g., including experimental results on the Heartsteps task by defining different risk levels through binning by number of steps in the prior 40 minutes).
4. What does “arbitrarily” mean in the context of line 89, right column? Does it truly mean the risk distribution is treated as random as a function of time, or can the problem formulation be treated as an MDP with treatments (or lack thereof) as the action space?
5. Is the $\rho$-robustness guarantee in lines 199-202 achievable in practice for non-trivial values of $\rho$? I would imagine that achieving such a bound for arbitrarily inaccurate estimates for $\tau^*$ is challenging if not impossible.
6. In line 371, right column, is there a clinical motivation for the choice of 150 steps to define the risk variable? Perhaps a relevant citation(s) or additional discussion would help support this choice.

**Questions For Authors:**

Please see my main numbered comments in the **Theoretical Claims** and **Other Strengths and Weaknesses** sections above. If the authors are able to address my numbered comments [1], [3], and [5] above, I would be happy to increase my score recommendation. Comments [4] and [6] are helpful for potential discussion (whether only in the discussion period or in the final manuscript as well), but would likely not change my evaluation of the paper.

**Relation To Broader Scientific Literature:**

The idea of OUS as a field is interesting to me, and I agree with the authors that prior work in this field is sparse given my (frankly limited) experience in the digital health-ML literature. While the algorithms introduced by the authors are notable, I think one of the key contributions of this work is the authors' principled formulation of OUS as an online optimization problem, and defining relevant metrics and definitions to characterize algorithms in this space.

**Theoretical Claims:**

I checked for the correctness of the Proofs in Appendix C, and have the following follow-up question:

1. In the equations presented on pages 11 and 12 regarding the proofs for Subroutines 1 and 2, the authors make the following approximations:
  $$f(b):=b-\frac{b}{e-1}\log(e-1)+\frac{b}{e}\approx b$$
  $$g(b):=\frac{2b}{e}+\frac{b}{e-1}-\frac{2b}{e^2} \approx b$$
where I define $f(b), g(b)$ for convenience of discussion. It seems that both $f(b), g(b) > b$ for all $b > 0$. Wouldn't this mean that the bound on the expectation budget may not be upper-bounded by $b$ even if it is upper bounded by $f(b), g(b)$?

---

> ### Author Rebuttal · Authors · 2025-04-01
>
> We are glad that the reviewer found our paper "well-written with notable clarity and presentation." We thank the reviewer for the valuable feedback and respond them in detail below.
> - **Proof of Lemmas 3.1 and 4.1** Sorry for the sloppiness in the proofs. To ensure exact satisfaction of $\mathbb{E}\left[\sum_{i=1}^{\tau^*} p_{i}\right] \leq b$, one can appropriately scale the input budget $b$ in Algorithms 1 and 2. For example, for Subroutine 2, the input budget should be set as $b/1.047$. In our experiments, we have implemented this adjustment to guarantee that the budget constraint is strictly satisfied. We will revise the statement of the lemmas and theorems to reflect this.
> - **Multiple risk levels** Binary or categorical risk levels are common in digital health studies. For instance, the HeartSteps study defines risk levels as binary, with fewer than 150 steps in the previous 40 minutes considered "at risk", and otherwise "not at risk".  Similarly, in the Sense2Stop Smoking Cessation Study [2], the risk variable takes on three categories: stress, no stress, and unknown.  Each distinct risk category typically requires a separate budget input, resulting in independent subproblems.
> In a truly continuous risk setting, the probability of observing any specific risk value is effectively zero, making the OUS formulation unsuitable. However, a pseudo-continuous setting is feasible by discretizing risk into  different levels and applying our proposed methods separately for each category, as described in Appendix A. We provide empirical results demonstrating this approach using the HeartSteps data, categorizing step counts into three levels: fewer than 50, fewer than 100, and fewer than 150 steps. The results suggested that our proposed learning-augmented algorithm in general has the best performance. Figures illustrating these results are available at the following link: https://imgur.com/a/UXRotSQ
> - **l89, arbitrarily** By "arbitrarily," we mean that the distribution of risk levels can change adversarially over time without assuming an underlying structure like an MDP. Since interventions may unpredictably influence future risks, our algorithm is designed to ensure strong performance guarantees even in the worst case.  We will clarify this in l89.
> - **$\rho$-robustness** Thank you for the question. Lines 119–122 explicitly state the conditions for achieving $\rho$-robustness. In maximization problems like OUS, the goal is to design an approximation algorithm with the highest possible robustness factor $\rho$. Theorems 3.2 and 4.2 specify the robustness factors that our algorithm can achieve, demonstrating satisfactory performance. Non-trivial robustness levels, such as $\rho > 1-\frac{1}{e}$ are typically challenging or impossible to achieve for maximization problems, as supported by existing literature [3].
> - **clinical motivation for risk variable** Thank you for raising this. Yes, previous clinical studies define sedentary behavior as inactivity having at least 40 minutes of time with fewer than 150 steps. We will include additional reference [1] to reflect this.
> - **x-axis of Figure 3** Thank you for the question. The x-axis correctly represents "Width," not user-days, as the average competitive ratio shown is computed across user-days for varying widths of the prediction interval. We display results across different confidence interval widths to assess algorithm performance under predictions ranging from extremely good to extremely bad, thereby testing their robustness in practice.
>
> Thank you for catching the typo. We have corrected it.
> - [1] Spruijt-Metz, et al. (2022). Advancing behavioral intervention and theory development for mobile health: the HeartSteps II protocol. International journal of environmental research and public health.
>
> - [2] Battalio, S. L., et al. (2021). Sense2Stop: a micro-randomized trial using wearable sensors to optimize a just-in-time-adaptive stress management intervention for smoking relapse prevention. Contemporary Clinical Trials.
>
> - [3] Buchbinder, N., et al. (2007). Online primal-dual algorithms for maximizing ad-auctions revenue. In European Symposium on Algorithms.

---

> > ### Comment · Reviewer_Nntf · 2025-04-01
> >
> > I thank the authors for their thoughtful rebuttal and hard work. All of my initial concerns raised in my initial review have been addressed/will be addressed in the camera-ready version, and I will increase my score to 4. However, I do note that I do not regularly do research in this space and acknowledge that my review may have missed more niche/domain-specific points - for this, I defer to the expertise of more knowledgeable reviewers.

---

### Official Review · Reviewer_7RoC · 2025-02-26

**Overall Recommendation:** 3

**Summary:**

This paper studies the following online prediction problem: Let $\tau^*$ be an unknown number in $[b,T]$, where $b$ and $T$ are known. At every step $t\le \tau^*$, the learner needs to predict a number $p_t\in [0,1]$. The goal is to maximize
$
\sum_{t=1}^{\tau^*}p_t-\frac{1}{\tau^*}\ln\left(\frac{\max_t p_t}{\min_t p_t}\right)
$
subject to the constraint
$
\mathbb{E}\Big[\sum_{t=1}^{\tau^*}p_t\Big]\le b,
$
where $\mathbb{E}$ denotes the expectation with respect to the internal randomness of the prediction algorithm.

The paper provides an algorithm that achieves a constant competitive ratio for the above objective compared to the optimal offline solution (where $\tau^*$ is known). The paper further provides an argument demonstrating that the competitive ratio cannot be better than $0.504$. Finally, the paper considers the case when $\tau^*$ can be predicted within a certain interval and demostrate the ultility of the approach in the context digital health.

**Claims And Evidence:**

First of all, the problem formulation seems quite problematic since the term
$
\frac{1}{\tau^*}\ln\left(\frac{\max_t p_t}{\min_t p_t}\right)
$
can be negligible even if $p_t$ decreases polynomially with respect to $t$. One can simply assign $p_t$ to be any convergent series, such as
$
p_t \propto \frac{1}{t^2}.
$
In this case, the competitive ratio can easily be made to approach $1$, provided that $\tau^* \gg b$. This suggests that the proposed problem might be essentially trivial.

The authors also claim that no algorithm can achieve a competitive ratio better than $0.504$. However, I find the proof problematic. Specifically, how can one assume that the optimal strategy must be decreasing? While I understand that this should be the case for the "sum part" via the rearrangement inequality, what about the "log penalty" part?

**Essential References Not Discussed:**

N/A

**Experimental Designs Or Analyses:**

The paper conducts some evaluations using synthetic and real data, which look reasonable to me. However, since I am not familiar with the compared benchmarks, I cannot comment on the significance of the experimental results.

**Methods And Evaluation Criteria:**

As far as I understand, the proposed method is essentially a "doubling trick", which is widely used in the online learning literature to obtain time-independent guarantees, with some tweaks to the specific parameter regime.

**Other Comments Or Suggestions:**

The $\sigma$ that first appears on page 3 (line 139, right) is not defined.

**Other Strengths And Weaknesses:**

**Other Weaknesses:**

1. The problem studied is more of an algorithmic design problem rather than a learning problem. From a machine learning perspective, the problem introduced is quite trivial, as the environment provides nearly no feedback (except for termination). Therefore, I believe the paper might not quite fit within the scope of ICML.

2. The paper provides no intuition behind the design of the algorithms. For example, why choose the distribution proportional to $1/\alpha$, and why choose the constant $e$? My impression is that the selection of such parameters appears quite arbitrary, and the authors did not put enough effort into optimizing and justifying these choices.

**Questions For Authors:**

Please answer the questions in **"Claims and Evidence"** and address the concerns in **"Other Strengths and Weaknesses."**

**Relation To Broader Scientific Literature:**

As far as I understand, the primary contribution appears to be providing a mathematical formulation for a particular application scenario in digital health.

**Theoretical Claims:**

I verified some of the proofs, such as Lemma 3.1 and Theorem 3.2, and they appear to be correct (i.e., I did not identify any significant technical issues).

---

> ### Author Rebuttal · Authors · 2025-04-01
>
> We appreciate the reviewer's feedback. We clarify that our work is positioned as an applied paper motivated by digital health applications rather than a purely theoretical paper.  Below, we illustrate the type of guarantees that competitive ratios provide, detail the computation of competitive ratios, and explain how this work aligns with the ML community.
> - **is penalty term negligible** Consider your example where the series converges as $p_i \propto \frac{1}{i^2}$. Suppose $b=2$ and $\tau^*=4$. The resulting treatment probability sequence is $1$, $\frac{1}{4}$, $\frac{1}{9}$, $\frac{1}{16}$. Then, the penalty term becomes $\frac{1}{4}\ln \frac{1}{1/16} = \frac{\ln 16}{4}$. The objective function becomes $1+\frac{1}{4}+\frac{1}{9}+\frac{1}{16}-\frac{\ln 16}{4} \approx 0.73$. This yields a competitive ratio of approximately $\frac{0.73}{2} \approx 0.365$. Although as $\tau^*$ grows large, the penalty term $\frac{1}{\tau^*}\ln \frac{1}{1/{\tau^*}^2} = \frac{2\ln \tau^*}{\tau^*}$ becomes negligible, this does not imply that the competitive ratio approaches 1 in general.
> Since our problem makes *no* assumption on the distribution of $\tau^*$ and the competitive ratio must hold for the worst-case instance, the challenge arises precisely when $\tau^*$ is close to the budget $b$. Thus, the problem remains non-trivial.
> - **upper bound derivation** In our proof, we do not require the policy to be non increasing. Sorry for the typo, we have corrected it as below. To derive the upper bound of 0.504 using Yao's lemma, we constructed a challenging instance and solved the following maximization problem over probabilities $p_1, \dots, p_5 $:
>     \begin{align*}
> \arg \max_{p_1, p_2, p_3, p_4, p_5}\pi_1p_1 + \pi_2(p_1+p_2) + \pi_3(p_1+p_2+p_3) + \pi_4(p_1+p_2+p_3+p_4) + \pi_5 b - \pi_2\frac{1}{2} \left|\ln\frac{p_1}{p_2}\right|-\pi_3\frac{1}{3}\left|\ln\frac{p_1}{p_3}\right| - \pi_4\frac{1}{4}\left|\ln\frac{p_1}{p_4}\right| - \pi_5\frac{1}{5}\left|\ln\frac{p_1}{p_5}\right|
> \end{align*}
> subject to $p_1+p_2+p_3+p_4+p_5=b$.
> We mistakenly omitted the absolute value in the current proof. Notably, despite no explicit constraint enforcing a decreasing solution, the optimal solution naturally followed this pattern, suggesting that decreasing or non-increasing probabilities generally improve performance in the OUS problem.
> - **Doubling trick** In our problem formulation, there is no reward learning, so neither our method's inspiration nor our analysis techniques stem from online learning problems. While our algorithm design may resemble the doubling trick used in learning, the analysis objectives and problem complexities are fundamentally different.
> - **scope of ICML** We acknowledge that this paper focuses  on online optimization, specifically the design of randomized approximation and learning-augmented algorithms, rather than traditional online learning. Online optimization, especially with prediction augmentation, is actively studied in machine learning. Related works, such as [1] (ICML 2023) and [2] (NeurIPS 2020), demonstrate its relevance in top-tier conferences. Thus, we believe our paper aligns well with ICML's scope.
> - **parameter choices in algorithm design**
> We chose the distribution $\propto 1/\alpha$ and the constant $e$ for a cleaner analysis of the algorithm’s expected performance, particularly for integration. The choice of $e$ aligns with classical online maximization results where optimal competitive ratios often take the form $1 - 1/e$ [3]. It is further inspired by the randomized algorithm of [1].
> We have clarified this in the revised version (l208):
> >We choose the density $1/\alpha$ and the constant $e$ to simplify the analysis of the algorithm’s expected performance. The choice of $e$ is also motivated by its frequent appearance in upper bounds for online optimization problems.
> - **l139, $\sigma$:** The tunable parameter $\sigma$ is introduced on l139 (right column), allowing us to explore a range of values that penalize non-uniformity at different scales. We have now revised l139:
> > The tunable parameter $\sigma$ in the penalty term $\sigma \cdot \ln \frac{\max_i p_i}{\min_i p_i}$ serves as a scaling factor to control the strength of the penalty. This allows flexibility in adjusting how strongly non-uniformity is penalized, as further discussed in Remarks 3.2 and 4.2.
>
> See our response to reviewer KP69 for more discussion on $\sigma$.
> - [1] Shin, Y., et al. Improved learning-augmented algorithms for the multi-option ski rental problem via best-possible competitive analysis. ICML, 2023.
>
> - [2] Bamas, E., et al. The primal-dual method for learning augmented algorithms. NeurIPS, 2020.
>
> - [3] Buchbinder, N., et al. Online primal-dual algorithms for maximizing ad-auctions revenue.  European Symposium on Algorithms, 2007.

---

> > ### Comment · Reviewer_7RoC · 2025-04-03
> >
> > I thank the authors for the detailed response. I now understand your main contribution is to provide a competitive ratio for all the parameter region, which seems to be a solid one. Therefore, I increase the score to 3.

---

> > > ### Author Response · Authors · 2025-04-03
> > >
> > > We thank the reviewer for adjusting their score and would be grateful if they could also update the overall recommendation in the original review. Many thanks!

---

### Official Review · Reviewer_KP69 · 2025-03-05

**Overall Recommendation:** 2

**Summary:**

This paper investigates the problem of online uniform sampling (OUS), where the goal is to allocate a budget uniformly across unknown decision times. The authors formulate the OUS problem as an online optimization problem and propose randomized algorithms to address it. To evaluate the performance, they consider competitive ratio, consistency, and robustness. The proposed method outperforms previous approaches on real-world data.

---

### update after rebuttal and discussion period

After the rebuttal, I raised my score from 2 to 3, taking into account the positive comments from other reviewers, as I am not familiar enough with the topic to fully assess the contribution and novelty of the paper. However, during the discussion period, I became increasingly skeptical about the problem setting itself. While the authors claim that $\tau^*$ is revealed to the algorithm online, it appears that the only observable information is whether $\tau^*$ has been reached. Moreover, the proposed algorithms seem capable of constructing $p_i$ in advance, without any interaction with the environment, which suggests that the problem is an offline setting. For these reasons, I have decided to maintain my original score.

**Claims And Evidence:**

The general claims appear to be reasonable.

**Essential References Not Discussed:**

I do not have specific idea.

**Experimental Designs Or Analyses:**

The general claims appear to be reasonable, with one exception.

In the synthetic experiments, the authors provide results for their proposed methods and a naive benchmark algorithm. However, since the learning-augmented algorithm requires a confidence interval as input, one could provide a naive point estimate $\tau^*$ (e.g., $(U+L)/2$) for the heuristic algorithm by Liao et al. (2018). Is there a specific reason why the results from Liao et al. (2018) are omitted in the synthetic experiments, whereas they are included for the HeartSteps V1 dataset?

**Methods And Evaluation Criteria:**

The general claims appear to be reasonable, with one exception.

Why we need to consider/define $\lambda$-consistency of algorithms?
If the algorithms are designed to solve OUS problem, it should be $1$-consistent as the definition assume the perfect prediction.

**Other Comments Or Suggestions:**

### Suggestion

- Using the same acronym for online uniform sampling and online uniformity scheduling (in Section 5) is confusing. Since the latter is not used frequently, it would be better to remove it.

- When the proof relies on the increasing/decreasing property of a function, it would be better to rewrite the equations explicitly to make these properties clear. For example, it is not straightforward to verify in L703 and L898.

- The current lines in figures are thin.

**Other Strengths And Weaknesses:**

### Strength

- The first randomized algorithm to solve OUS problem by formalizing OUS problem as an online optimization problem.

**Questions For Authors:**

Q1. See Methods And Evaluation Criteria

Q2, Q3. See Theoretical Claim section

Q4. See, Experimental Designs Or Analyses section

Q5. What is the role of randomness of $\text{Int}\tilde{\tau}$ in algorithms? Its value can change at each time step, but by at most 1.

**Relation To Broader Scientific Literature:**

I do not have specific idea.

**Theoretical Claims:**

I checked proofs in appendix while I skipped some calculation details.
The followings are my claim on the results of the paper.

### Claim 1. Incomplete Proofs for Lemma 3.1 and Lemma 4.1

Lemmas 3.1 and 4.1 claim that the expected sum of probabilities is bounded by the budget constraint $b$.
However, upon examining the proofs in the appendix, it appears that the authors rely on the following approximation (L643):
$$
b-\frac{b}{e-1}\ln(e-1)+\frac{b}{e} \approx b.
$$
Explicit computation, however, shows that:
$$
b-\frac{b}{e-1}\ln(e-1)+\frac{b}{e} \approx 1.05b > b.
$$
This discrepancy suggests that the current proof does not fully establish Lemma 3.1.
To further investigate, I checked explicit values where similar approximations were used:

- Subroutine 2 (L676): the value is $1.047b$.
- Subroutine 3 is fine.
- Subroutine 4 (L973): the value is $1.07b$.
- Subroutine 5 (L1020): the value is $1.018b$.
- Subroutine 6 (L1085): it seems it depends on the value of $L$, where I am not sure.

Based on these findings, I argue that the current proof does not support the correctness of Lemmas 3.1 and 4.1. A more careful analysis is required to ensure that the budget constraint holds.

---

### Claim 2. Remark 3.3: Conditions on $\sigma$

The proof of Theorem 3.1 introduces $\sigma$ without a clear explanation (though it is mentioned briefly in the main text). In Subroutine 2, the authors argue that the last inequality in L781 holds because the function is increasing with respect to $\beta$.
However, this is only true under the condition, $\sigma \leq b/e$.
Similarly, in Subroutine 3, the same technique is used, which holds only if: $\sigma \leq b/e-b/e^2$.
This follows from the equation (L912):
$$
\frac{b}{e} - \frac{\beta}{e^2} + \frac{\beta}{e^2}  - \frac{b}{e^2} +  \left(\frac{b}{e} - \frac{b}{e^2} \right)\ln \frac{\beta}{b} -\sigma\ln \frac{\beta}{b} =\frac{b}{e} - \frac{b}{e^2} + \left(\frac{b}{e} - \frac{b}{e^2} -\sigma \right)\ln \frac{\beta}{b}.
$$
For the function to be increasing w.r.t. $\beta\in [b,be]$, the coefficient must be positive, which imposes the condition on $\sigma$.
Thus, it would be beneficial to clarify these constraints explicitly in the proof and revise the description in Remark 3.3.

---

> ### Author Rebuttal · Authors · 2025-04-01
>
> We appreciate the reviewer's feedback. Below we address each point in detail to further clarify and strengthen the paper. We have also included SeqRTS as a benchmark in the synthetic experiments. We would be happy to provide further clarification if needed.
> - **The need for consistency**  While perfect predictions ($U = L = \tau^*$) could theoretically achieve $\text{OPT}(\tau^*)$, most learning-augmented algorithms sacrifice 1-consistency to maintain robust across prediction uncertainties [1,2]. Regardless of the confidence interval width, the algorithm must ensure an acceptable worst-case guarantee. Although a separate algorithm could be designed for perfect predictions, consistency measures performance as prediction accuracy improves. In contrast, our algorithm is designed to achieve 1-consistency.
> - **Proof of Lemmas 3.1 and 4.1**  Sorry for the sloppiness in the proofs. To ensure exact satisfaction of $\mathbb{E}\left[\sum_{i=1}^{\tau^*} p_{i}\right] \leq b$, one can appropriately scale the input budget $b$ in Algorithms 1 and 2. For example, for Subroutine 2, the input budget should be set as $b/1.047$. In our experiments, we have implemented this adjustment to guarantee that the budget constraint is strictly satisfied. We will revise the statement of the lemmas and theorems to reflect this.
> - **$\sigma$ in proof of Theorem 3.1** We have revised the text at line 139, c2 to clarify that $\sigma$ acts as a tunable parameter in the penalty term $\sigma \cdot \ln \frac{\max_i p_i}{\min_i p_i}$, scaling the penalty strength to flexibly adjust non-uniformity penalization. We have revised the proof and modified the description in Remark 3.3 accordingly to incorporate the condition on $\sigma$:
> >In Appendix C.2, we show that for Subroutine 1, Theorem 3.2 holds over a wide range of $\sigma$ values, specifically $\sigma \leq \frac{2b(\ln(e-1)-1+1/(e-1))}{\ln b+2-2\ln(e-1)}$.
> For Subroutine 2, Theorem 3.2 remains valid when $\sigma \leq \frac{b}{e}$. For subroutine 3 where $\tau^*$ can be unbounded, the theorem holds for $\sigma \leq \frac{1}{2-\ln(e-1)} \frac{1}{e}\left(1-\frac{1}{e}\right)^{j^*+1} b$, ensuring that the penalty term scales similarly to the budget term in the objective.
>
> We clarify that for subroutine 3, only requiring $\sigma \leq b/e-b/e^2$ is not enough for the case where $j^*\geq1$. The stronger condition is necessary to maintain a valid competitive ratio via the monotonic property; otherwise, the competitive ratio may become arbitrarily small.
> - **Including SeqRTS on synthetic data** Thank you for the suggestion. We have now conducted additional synthetic experiments using the SeqRTs algorithm proposed by Liao et al. (2018) with the suggested naive point estimate $(U+L)/2$. The results are provided on  https://imgur.com/a/Vs7nDAu. The results suggest that SeqRTS, in general, performs worse compared to our algorithms.
> - **The role of randomness of Int $\tilde{\tau}$ in algorithms** Taking Algorithm 1 as an example, $\text{Int} \tilde{\tau}$ determines whether the current risk time $i$ exceeds the length of the current stage, thereby indicating whether the budget should be updated.  Because stage lengths must be integer-valued, $\tilde{\tau}$ must be rounded. To avoid rounding error, we adopt a stochastic rounding method, ensuring that the expectation of the rounded stage length, $\text{Int} \tilde{\tau}$, matches exactly the original value $\tilde{\tau}$. We will further clarify this in our revised paper.
>
> Thank you for catching the typo on online uniform scheduling and suggestions on the figure. We have revised our paper accordingly. Additionally, we will explicitly explain in our proofs the steps that rely on the monotonicity of the functions. Thank you for the suggestion.
>
> - [1] Bamas, E., Maggiori, A., \& Svensson, O. (2020). The primal-dual method for learning augmented algorithms. NeurIPS.
>
> - [2] Kevi, E., \& Nguyen, K. T. (2023). Primal-dual algorithms with predictions for online bounded allocation and ad-auctions problems. International Conference on Algorithmic Learning Theory.

---

> > ### Comment · Reviewer_KP69 · 2025-04-04
> >
> > I appreciate the authors' efforts in addressing my concerns, particularly the theoretical analysis. Most of my initial questions and concerns have been resolved in the rebuttal.
> >
> > While I am not fully confident in assessing the contribution and novelty of this paper due to my limited knowledge to this field, I increased my score to 3, taking into account the comments from other reviewers.

---

### Official Review · Reviewer_FkxE · 2025-03-16

**Overall Recommendation:** 3

**Summary:**

The topic of this paper is online uniform sampling problem (OUS) - motivated by applications in digital health.

OUS problem is to distribute a sampling budget b uniformly across unknown decision times in horizon [1,T].  An adversary chooses a value tau* in interval [b,T], revealed only online.  At each decision time i ∈ [τ*], the goal of the algorithm is to determine a sampling probability to (i) maximize budget spent and (ii) achieve a distribution over tau* that is as uniform as possible.

The paper obtains a randomized algorithm for OUS; it also extends it to incorporate predictions (intervals for tau*), say, LA-OUS (for learning-augmented).  The paper shows LA-OUS is consistent and robust.

There are synthetic data experiments showing the performance of OUS and LA-OUS.  There is also an experiment on HeartSteps mobile application, which also shows the algorithms work well.

**Claims And Evidence:**

Looks good.

**Essential References Not Discussed:**

No.

**Experimental Designs Or Analyses:**

Looks good.  One real-world dataset is a bit unsatisfactory but it is conceivable that the datasets are limited for this problem.

**Methods And Evaluation Criteria:**

Looks good.

**Other Comments Or Suggestions:**

The model has to be spelled out precisely: for each time i in [1, T], what happens and what is the response.

l118 c1: explain ", which is revealed to the algorithm online."

153 c1: why is "penalizing change in treatment probabilities within each level"  a useful constraint wrt the "uniform" objective?

The second term in the optimization problem in l153 - why not measure the L1 distance to uniform? That is \sum_i | p_i - b/\tau^*| - Why this particular objective?

l110 c2: what is the randomness in the algorithm?

l204 c1: for consistency, what is the smooth tradeoff on the competitive ratio, as a function of (U-L)?

l229 c1: is j maintained as state in subroutine 2/3 - please clarify?

Is there a scenario where it does not make sense to utilize all budget?

line 332 col 2: typo (in Algorithm 2).  similar typos in a few other places (eg, line 312 col 2)

line 321 col 2: what is the competitive ratio averaged over?

some text repeated between regular and learning-augmented setting - wonder if the text can be crisper

**Other Strengths And Weaknesses:**

Strengths

+ A principled approach to the OUS problem

+ Technically nice and non-trivial

Weaknesses

- Very niche application and not sure how broadly this problem occurs

- The lack of good upper bounds is a negative

- The case splitting resembles the one in Shin et al (ICML) - it will be good to clarify what is common and what is different.  I agree the settings seem different, but how come there is some similarity in the analysis/three regimes?

**Questions For Authors:**

Is it possible to generalize your algorithms to weighted risk times - each risk time i has a different weight w_i, with the definitions naturally extended to this setting?

Could you obtain stronger bounds if you are allowed to relax the budget a bit?  Some type of bicriteria setting.

Synthetic experiments: in addition to uniform at random choice for risk times, could you try some other distribution.  Eg, geometrically spaced at the beginning or at the end?

Is the three operating regime inherent or is it an analysis artifact?

**Relation To Broader Scientific Literature:**

The paper studies OUS through the lens of online algorithms and competitive analysis.

**Theoretical Claims:**

Looks good.

---

> ### Author Rebuttal · Authors · 2025-03-31
>
> We thank the reviewer for the positive review and are glad that the reviewer found our work as "a principled approach to the OUS problem and technically nice and non-trivial."
> - **Weighted risk times:** Weighting risk times differently implies varying risk levels, prioritizing higher-risk times. Our approach can naturally handle this by decomposing the problem into independent subproblems for each risk level, see Appendix A.
> - **Stronger bounds by relaxing budgeting:** Relaxing the budget constraint is unlikely to improve upper bounds for OUS. Our proof, using Yao's lemma, constructs a hard instance where all feasible deterministic algorithms perform poorly. Slightly relaxing the budget would not yield large improvements. A similar observation applies to the lower bound, where the budget already holds only in expectation. Further relaxation would make the constraint ineffective.
> - **Distribution of $\tau^\star$ in experiments:** Our proposed algorithms are, by design, independent of the specific distribution of risk times. They determine intervention probabilities solely based on the number of risk times $\tau^*$. Thus, for a fixed $\tau^*$, the results remain the same whether the risk times are uniformly distributed or geometrically spaced.
> - **Case splitting compared with Shin et al/three regime inherent or analysis artifact?** While Shin et al. proposed a single unified randomized algorithm without different regimes, providing guarantees as $T\rightarrow\infty$, we focus on finite-horizon guarantees under the additional budget constraint $b$. For example, our Algorithm 1 considers three regimes: 1) $T \leq be$ (Subrt 1), 2) $be < T \leq be^2$ (Subrt 2), and 3) $T > be^2$ (Subrt 3). Case 3 resembles Shin et al.'s infinite-horizon setting, where both $T$ and $\tau^*$ can be unbounded. However, our budget constraint introduces new challenges, necessitating novel algorithm design. In Case 3, to meet the budget constraint, the algorithm must initially adopt a conservative approach, resulting in a slightly lower competitive ratio. To mitigate this, we introduce specialized algorithms (Subrts 1 and 2) for the first two regimes, achieving higher competitive ratios. Conversely, both Shin et al.'s *proof* and ours rely on a case-by-case analysis to evaluate the robustness of the learning-augmented algorithm, as computing the expected cost depends on identifying the phase in which the algorithm terminates. We will add the above clarification to our revised paper.
> - **Model must be spelled out precisely:**  We revised L81 c2 as follows:
> >At each decision point $t\in[1, T]$, the algorithm observes the binary risk level $R_t$ associated with the patient. Here, $R_t=1$ indicates a heightened likelihood of an adverse event, such as a relapse into smoking, while $R_t=0$ implies a lower risk level.
> - **l118 c1** We revised L93 c2 as follows:
> >Since $R_t$ is revealed to the algorithm online, the total number of risk times throughout the decision period, $\tau^* = \sum_{t=1}^T R_t$, remains *unknown* until the last decision point $T$.
> - **153 c1 L1 distance in penalty term:** Our choice of penalty ties to the uniform constraint through the two primary objectives outlined in Section 2.1 L122-124. Other alternative penalty terms that satisfy the properties described in Remark 2.1 could also be considered. However, the proposed L1 distance penalty is not appropriate, as it does not sufficiently penalize cases where probabilities are zero ($p_i=0$), which can result in budget depletion before the final risk time.
> - **l110 c2** The randomness of the algorithm comes from the initialization step, where $\alpha\in [b,be]$ is sampled from a distribution with probability density function $f(\alpha) = 1/\alpha$. This sampled value $\alpha$ is subsequently used to calculate treatment probabilities. We have further clarified this point on L210.
> - **l204 c1 consistency tradeoff:** Consistency, as defined in L195, is defined relative to a perfect prediction of the risk times, i.e., when $U=L=\tau^*$. Thus, there is no trade-off between consistency and prediction interval width $U-L$. Our algorithm achieves 1-consistency. Further discussion on consistency is included in our response to Reviewer KP69.
> - **l229 c1 on j:** $j$ acts as a stage counter, incrementing when $i > \text{Int}\tilde{\tau}$ and triggering a budget update when $j>3$ to avoid budget depletion before the final risk time. For each problem instance is assigned to a specific subroutine based on $T$, separate counters $j_1$ and $j_2$ for Subrts 2 and 3 are unnecessary.
> - **Not using all budget:** While full budget utilization is ideal for inference, given that $\tau^*$ is random, achieving this under uniform constraints is hard in practice.
> - **l321 c2** We revised l321c2 as follows:
> > The evaluation metric is the average competitive ratio computed from 500 experimental replications.
>
> Thank you for catching the typo and redundant texts. These have been fixed.

---

### Decision · Program_Chairs · 2025-05-01

**Decision:**

Reject

**Comment:**

This paper studies the problem of deciding online a sampling distribution that aims to be close to uniform and uses its whole budget.
The authors propose a regularized objective which is maximized for uniform distributions. They present an algorithm that computes a sampling distribution and prove a bound on the competitive ratio.

My first concern is that this is not really an online learning problem. The decision maker does not receive any actionable information while the game proceeds, instead it is an offline optimization problem to maximize the worst-case objective.

My second concern is that the proposed bounds are almost constant with a minor dependency on the known problem parameters b and T. While for b=1, a constant competitive ratio is optimal, it does seem to converge to 1 as b increases. The bound of the authors completely fails to capture this.

My third objective is that it is very unclear that the regularization and problem formulation leads to useful distribution schemes. A quick analysis shows that the optimal strategy is using almost all budget initially and dividing the remaining budget with exponential decay over the remaining time-steps. This gives the max min solution, but does not seem to be a desirable plan from the real world problem formulation.